# Revealing the role of ionic liquids in promoting fuel cell catalysts reactivity and durability

Arezoo Avid[1,2], Jesus López Ochoa[1,2], Ying Huang [2,3], Yuanchao Liu [1,2], Plamen Atanassov [1,2] & Iryna V. Zenyuk [1,2,3] ✉

Ionic liquids (ILs) have shown to be promising additives to the catalyst layer to enhance oxygen reduction reaction in polymer electrolyte fuel cells. However, fundamental understanding of their role in complex catalyst layers in practically relevant membrane electrode assembly environment is needed for rational design of highly durable and active platinum-based catalysts. Here we explore three imidazolium-derived ionic liquids, selected for their high proton conductivity and oxygen solubility, and incorporate them into high surface area carbon black support. Further, we establish a correlation between the physical properties and electrochemical performance of the ionic liquid-modified catalysts by providing direct evidence of ionic liquids role in altering hydrophilic/hydrophobic interactions within the catalyst layer interface. The resulting catalyst with optimized interface design achieved a high mass activity of 347 A g$^{-1}_{Pt}$ at 0.9 V under $H_2/O_2$, power density of 0.909 W cm$^{-2}$ under $H_2/air$ and 1.5 bar, and had only 0.11 V potential decrease at 0.8 A cm$^{-2}$ after 30 k accelerated stress test cycles. This performance stems from substantial enhancement in Pt utilization, which is buried inside the mesopores and is now accessible due to ILs addition.

Sluggish oxygen reduction reaction (ORR) occurring on the cathode catalyst layer of polymer electrolyte fuel cell (PEFC) generates a high overpotential that limits the power density of PEFC, requiring high loadings of scarce and precious platinum (Pt) electrocatalyst for the ORR[1]. Various strategies exist to develop high-performing cathode catalyst layers, these include nitrogen-functionalized supports[2–6] and modification of electrocatalysts with promising additives to enhance the catalytic activity[7–10] emerging as a key approach[11,12]. In this way, catalyst layers with ultra-low Pt loadings can be designed without relinquishing the long-term durability as cost and durability are the two major barriers for commercialization of fuel cell electric vehicles. Durability concerns become even more significant within the context of heavy-duty fuel cell vehicles because of more stringent efficiency and lifetime targets compared to light-duty vehicles[13]. United States

Department of Energy (DOE) has established the technical targets for Class 8 long-haul fuel cell trucks at 25,000 h lifetime, 68% peak efficiency, and $80/kW$_{net}$ fuel cell system cost by 2030[14].

It has been shown that the carbon support on which the Pt nanoparticles are dispersed plays a critical role on both stability and performance of the catalyst layer by affecting the porosity, corrosion rate, and mass transport properties[15]. High surface area (HSA) carbon support is considered to be a great candidate for fuel cell application owing to its high internal mesoporosity. It can host up to 75% of Pt active surface area within the wide mesopores with size between 5.5 and 14 nm. Pt nanoparticles present in the internal pores are not in contact with ionomer, reducing ionomer's sulfonate groups adsorption on Pt that generally impedes Pt's kinetic activity[16,17]. Although HSA Pt-based catalysts have substantial ORR activity, they tend to show

[1]Department of Chemical and Biomolecular Engineering, University of California Irvine, 221 Engineering Service Rd., Irvine, CA 92617, USA. [2]National Fuel Cell Research Center, University of California Irvine, 221 Engineering Service Rd., Irvine, CA 92617, USA. [3]Department of Materials Science and Engineering, University of California Irvine, 221 Engineering Service Rd., Irvine, CA 92617, USA. ✉e-mail: iryna.zenyuk@uci.edu

high proton resistance within the catalyst layer's ionic network. This along with high oxygen mass transport resistance contribute to significant voltage loss, specifically at high current densities, where local transport restriction become more pronounced and more protons and oxygen are required to drive the ORR reaction[18,19]. Because the ionomer cannot penetrate the micropores and smaller mesopores of HSA carbon supports due to size-exclusion[20], liquid water is responsible for proton transport to Pt reaction sites[21]. However, water's ionic conductivity is orders of magnitude smaller than ionomer and even in confined environments it is two orders of magnitude smaller than that of Nafion[22]. Therefore, filling the micropores and smaller mesopores with ionic liquids (ILs) with high ionic strength in a range between $(2.77–5.3 \times 10^3)$ mol m$^{-3}$ is an effective approach to provide sufficient

proton delivery to the Pt active sites within the pores (Fig. 1a)[23,24]. ILs within the micropores and smaller mesopores induce local nanoconfinement that can promote ORR because: (1) Knudsen diffusion of oxygen taking place in the nanopores increases the frequency of oxygen molecules collision with the catalyst surface sites rather than molecule-to-molecule collision, (2) electric double layers (EDLs) are thin for ILs|Pt interface compared to water|Pt interface[24]. At potentials above 0.23 V, Pt is positively charged and will expel protons from the micropores and mesopores if in contact with water because double layers are thick. However, when Pt is in contact with ILs, the ionic strength of ILs is high and therefore the double layers will be thin, thus; electric field will be shielded within <0.5 nm from the surface of Pt and protons will not be impacted by positively charged Pt[24–26].

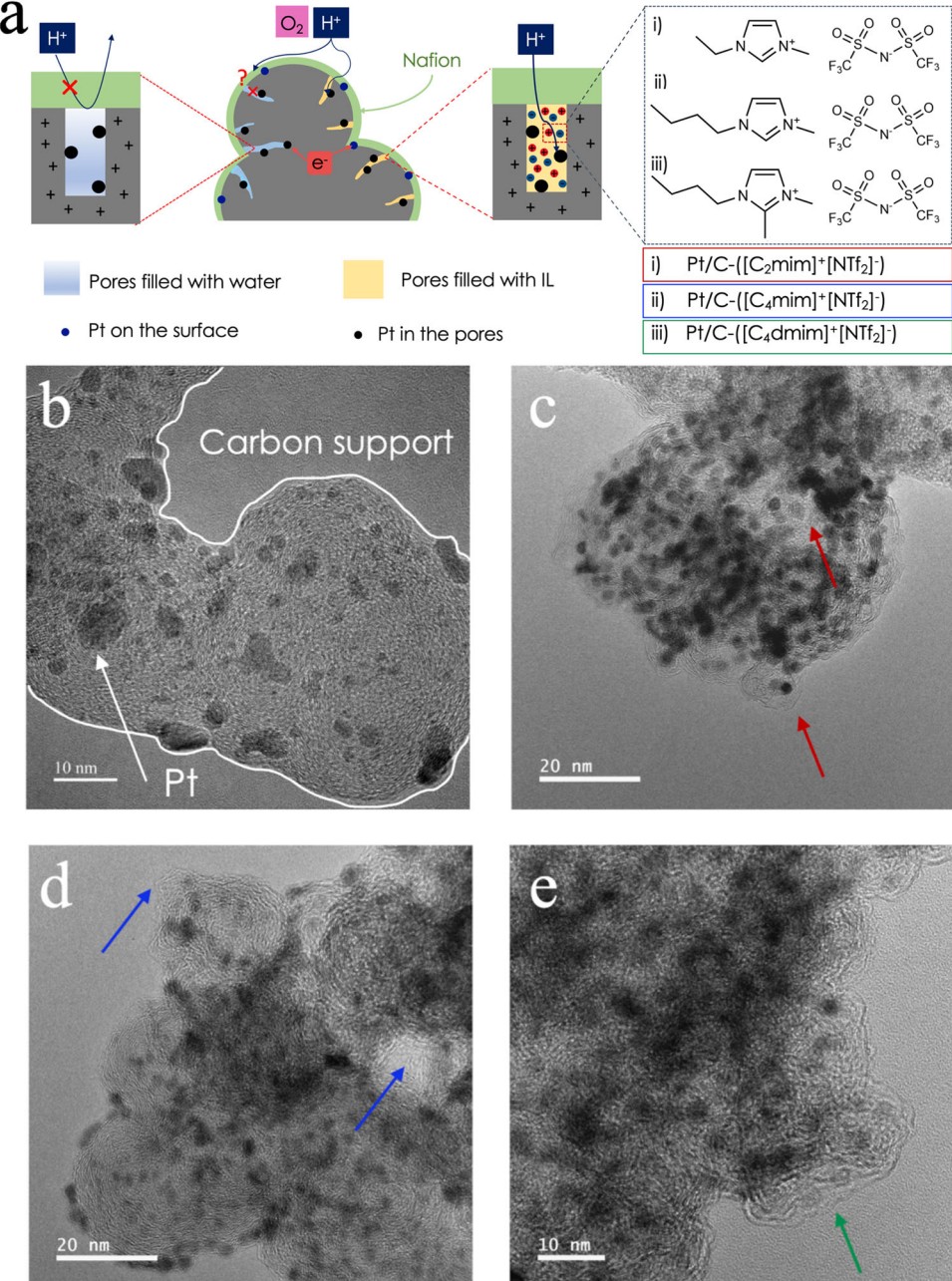

**Fig. 1 | Schematic drawing of catalyst layer interface and TEM images of pristine and modified catalyst powders. a** HSA Pt/C catalysts with micropores and smaller mesopores filled with either water or imidazolium-derived ILs, TEM images of **b** pristine 40 wt% HSA Ketjenblack Pt/C, **c** Pt/C modified with 1-ethyl-3-methylimidazolium bis(trifluoromethylsulfonyl)imide (Pt/C-([C$_2$mim]$^+$[NTf$_2$]$^-$)),

**d** Pt/C modified with 1-butyl-3-methylimidazolium bis(trifluoromethylsulfonyl) imide (Pt/C-([C$_4$mim]$^+$[NTf$_2$]$^-$)), and **e** Pt/C modified with 1-butyl-2,3-dimethylimidazolium bis(trifluoromethylsulfonyl)imide (Pt/C-([C$_4$dmim]$^+$ [NTf$_2$]$^-$)). Arrows show the ionic liquids lumps.

Various research groups have incorporated IL small molecules or IL-containing ionomers into the cathode catalyst layers and showed improved catalytic activity toward ORR[8,9,27,28], which was most often attributed solely to higher oxygen solubility and suppressed adsorption of non-reactive oxygenated species. Recently, Huang et al.[29] and Yan et al.[30] indicated that sufficient proton conductivity in the catalyst layer can be responsible for facilitated ORR kinetics. They also have shown that increased solubility of oxygen in ILs domains is compensated by poor diffusivity, as it is the product of these two that guides oxygen transport. However, they only relied on rotating disk electrode (RDE) studies to prove their hypothesis without translating it into membrane electrode assembly (MEA) fabrication and practical testing in fuel cell hardware.

In this work, we investigate in-depth electrochemical and physical properties of ILs impregnated into HSA Pt/C catalyst and integrate these catalysts into MEAs to observe their behavior under fuel cell operation. Alkyl imidazolium bis-(trifluoromethylsulfonyl)imide ILs have met the criteria of suitable modifier for fuel cell catalyst design due to their high ionic conductivity and oxygen solubility, low melting point, viscosity, and vapor pressure, and superior thermal and electrochemical stability under fuel cell tested temperature and voltage. They are also non-flammable and have high $\Delta pK_a$ value which is shown to be sufficient for satisfactory proton transfer[24,31–33]. It has been shown that the $pK_a$ of the cations in the IL dictates the local proton activity near Pt active sites. Technically, by the formation of hydrogen-bond network at the Pt|IL interface, the cation of IL would function as a proton donor facilitating the kinetics of the reaction[34]. Catalytic activity metrics including mass (MA) and specific activities (SA) of Pt/C-IL systems with optimized amount of IL were first pre-screened by RDE setup. Electrochemical impedance spectroscopy (EIS) was implemented to measure overall effective proton diffusion resistance ($R_H^+$) within IL-containing catalyst layer[29]. Then, MEAs with optimized ink recipes were fabricated and rigorously tested within PEFC hardware. The extensive in situ metrics including MA, SA, electrochemical surface area (ECSA), H2/air polarization curves, ionic conductivity determined by EIS, oxygen mass transport resistance, and CO displacement provided an insight into the effect of IL on the cell performance under harsh testing conditions. Ex situ characterizations such as transmission electron microscopy (TEM), X-ray photoelectron spectroscopy (XPS), nitrogen physisorption, zeta potential and water uptake measurements were also performed to determine the physiochemical properties and microstructure of the catalyst layers. Additionally, these characterizations helped understand the fundamental interaction between IL molecules and Pt surface, as well as the underlying mechanism for performance improvement. Finally, the durability of the cells was assessed by accelerated stress tests (ASTs) including potential cycling between two potentials to induce accelerated degradation of the catalyst layer[35]. This study, brought the understanding of catalyst layer integration with ILs in which not only the loading of Pt can be decreased without exacerbating degradation, but also an improved ORR activity can be bundled with a superior durability.

## Results and discussion
### Physical characterization of IL-modified samples
This study is focused on three types of ILs: 1-ethyl-3-methylimidazolium bis(trifluoromethylsulfonyl)imide ([C2mim]+[NTf2]−), 1-butyl-3-methylimidazolium bis(trifluoromethylsulfonyl)imide ([C4mim]+[NTf2]−), and 1-butyl-2,3-dimethylimidazolium bis(trifluoromethylsulfonyl)imide ([C4dmim]+ [NTf2]−). These imidazolium-derived ILs were selected because of their favorable ionicity, transport properties, and stability. The anion structure of all three ILs is the same, but the cation is varied with the length of the alkyl chain. The difference between 1-butyl-3-methylimidazolium bis(trifluoromethylsulfonyl)imide and 1-butyl-2,3-dimethylimidazolium

bis(trifluoromethylsulfonyl)imide is that the latter one has an additional methyl group in its cation chemical structure. Here we want to investigate the effect of various alkyl chain length and the additional anchoring methyl group on the physical and electrochemical properties of the catalyst. Figures 1b–e and S1 show the TEM images of HSA Ketjenblack Pt/C catalysts modified with these three ILs. Figure 2a demonstrates the nitrogen physisorption isotherms for baseline Pt/C and all IL-modified samples with IL/C of 1.28, at 77 K. Unlike Pt/C baseline isotherm with a significant nitrogen uptake of ~100 cm³ g⁻¹, all three IL-modified samples showed near zero nitrogen uptake at relative pressure below 0.01 (P/P0 < 0.01). However, at high relative pressure (P/P0 > 0.9), their sorption isotherm curves showed similar shape to that of the pristine Pt/C sample as indicated in Fig. 2a. As illustrated by the log-scale isotherm in Fig. 2a, the nitrogen uptake starting points for IL-modified samples were around 0.01 P/P0, indicating a complete filling or blocking of sub-micropores below 1 nm. Pore size distribution (PSD) as a function of pore diameter is shown in Fig. 2b in order to assess the filling degree and IL coverage of the IL-containing Pt/C powders. For pores with diameter above 50 nm, almost no difference was observed between the pore volume of the baseline and three IL-modified samples indicating that the macropores of the modified powders are not filled with ILs. For pores between 2 to 50 nm, the pore volume decreased at least 50% showing that all IL molecules have partially filled mesopores. On the other hand, over 90% drop in the pore volume is observed for micropores between 1 and 2 nm indicating that the micropores are fully impregnated with ILs. This is in agreement with the significantly reduced nitrogen uptake below 0.2 P/P0 indicated in Fig. 2a. For the pore size between 2 nm and 20 nm, Pt/C-([C4dmim]+[NTf2]−) samples showed non-negligible difference as compared to the other two IL-modified samples (Fig. 2b), where its pore volume was still well below the Pt/C baseline but higher than that of its Pt/C-([C2mim]+[NTf2]−) and -([C4mim]+[NTf2]−) counterparts. This is partially due to the more complex structure of the [C4dmim]+, as it has both butyl and methyl cationic chains, higher free volume, and less packing of the molecules in the pores as a result. For interface modification in this work, small mesopores are of more relevance as Pt most likely cannot be deposited into micropores and the ionomer cannot access small mesopores below <20 nm. From BET observations and TEM images in Fig. 1, one can conclude that most of the Pt nanoparticles were well covered by the ILs.

Figure 2c shows the time evolution of water mass gain for baseline and IL-modified Pt/C coated films. By comparing the amount of water absorbed by the samples during the first 900 s while they are immersed into the water, it can be concluded that baseline Pt/C coated catalyst layer shows the least amount of water uptake of about 18 mg compared to all IL-containing catalyst layers. Higher amount of water adsorption in IL-containing catalyst layers indicates that the global hydrophilicity of Pt/C-IL-Nafion catalyst layers is more than that of the baseline. For instance, Pt/C-([C4dmim]+[NTf2]−)-Nafion catalyst layer has the highest water uptake amount of all (57 mg); hence, it is the most hydrophilic catalyst layer and has the highest global hydrophilicity. Here we can illustrate the potential intramolecular interactions between the IL molecules and Nafion domains with a schematic (Fig. S2). IL molecules are prone to create a local hydrophobic gradient near Pt surface and within the smaller mesopores that repels the SO3⁻ groups of Nafion to the outer surface resulting in more water absorption to the catalyst layer. Therefore, one can assume that the higher local/near-surface hydrophobicity is, the higher the global hydrophilicity of the whole the catalyst layer along with the Nafion would be. External contact angles shown in Table 1 were calculated using Wilhelmy equation and sudden increase in the water mass gain after each sample was removed from the water ($\triangle mg$). The details of experiment can be found in the Methods section of this paper. Pt/C-([C4dmim]+[NTf2]−)-Nafion film exhibited the smallest external contact angle of 137° and the highest amount of water uptake of 57 mg among

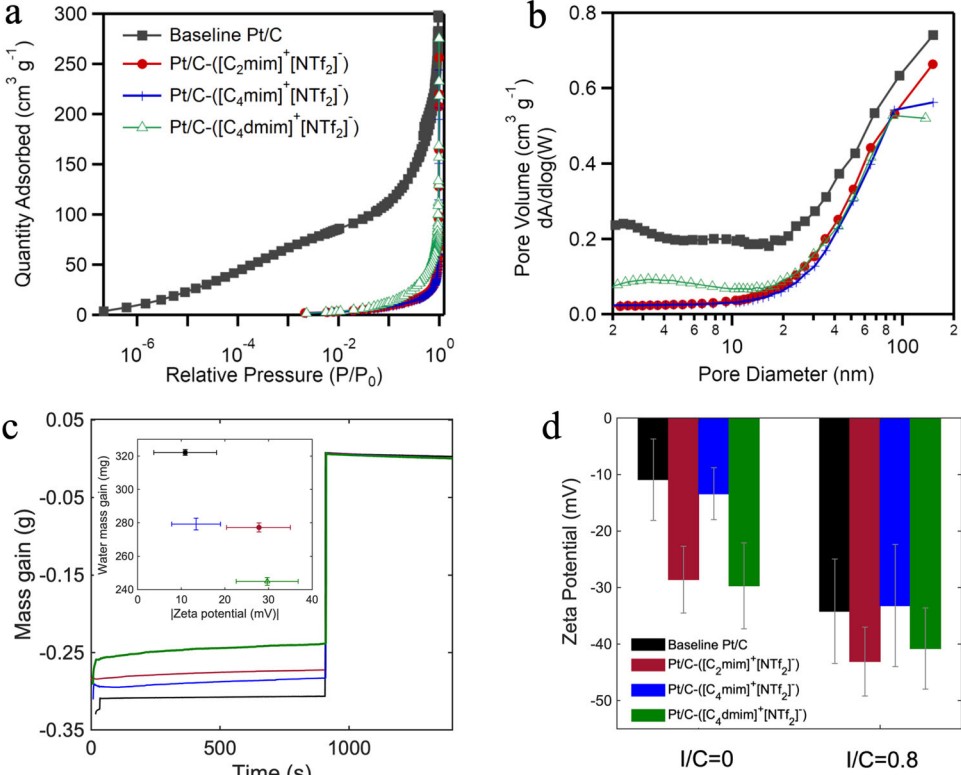

**Fig. 2 | Ex situ physiochemical characterization of IL-modified samples vs. baseline. a, b** Nitrogen physisorption analysis at 77 K for pore occupancy by three ILs: **a** Sorption isotherm with log-scale relative pressure to highlight the microporous region, **b** volumetric pore size distribution, **c** mass evolution of catalyst layers containing various ILs with IL/C ratio of 1.28 measured using capillary penetration method. **d** Zeta potential measurements for pristine and IL-modified Pt/C with and without Nafion (error bars represent standard deviation of the mean).

**Table 1 | Water uptake and external contact angle catalyst layers containing various ILs with IL/C ratio of 1.28 measured by Krüss tensiometer**

| Sample | Water uptake during first 900 s (mg) | △$mg$ at 900 s (mg) | External contact angle |
|---|---|---|---|
| Pt/C-Nafion | 18.2 ± 5.1 | 322.2 ± 1.8 | 164.3 ± 1.8 |
| Pt/C-($[C_2mim]^+[NTf_2]^-$)-Nafion | 34.2 ± 5.8 | 277.2 ± 2.7 | 147.3 ± 0.9 |
| Pt/C-($[C_4mim]^+[NTf_2]^-$)-Nafion | 33.1 ± 3.4 | 279.3 ± 3.5 | 148.0 ± 3.3 |
| Pt/C-($[C_4dmim]^+[NTf_2]^-$)-Nafion | 57.2 ± 6.8 | 244.9 ± 2.3 | 136.8 ± 0.7 |

all IL-containing samples. This indicates that ($[C_4dmim]^+[NTf_2]^-$) IL molecules create the most local/near-surface hydrophobic micro-environment and induce the strongest electrostatic repulsion on $SO_3^-$ groups of Nafion to reorient to the outer surface. These results are in agreement with zeta potential measurements in which local hydrophobic interactions between the IL molecules and Nafion's PTFE backbone possibly dictate the conformational structure of the Nafion polymer chains. A stronger electrostatic repulsion between the IL molecules and Nafion's PTFE backbone results in a higher local charge density at the interface, and larger zeta potential (Fig. 2c, d). From tensiometry results, we observed that Pt/C-($[C_4mim]^+[NTf_2]^-$)-Nafion catalyst layer has the largest external contact angel of 148° and the least water uptake of about 33 mg. Therefore, Pt/C-($[C_4mim]^+[NTf_2]^-$)-Nafion film is the most hydrophobic catalyst layer compared to the other two IL-containing catalyst layers. Hence, ($[C_4mim]^+[NTf_2]^-$) IL molecules create the least local/near-surface hydrophobic micro-environment compared to the other two ILs and induce weaker electrostatic repulsion on Nafion chains confirmed by zeta potential measurements.

Figure 2d and Table S1 represent the zeta potential of the suspensions containing Pt/C and IL-modified Pt/C in water/IPA solutions with 51 wt% water content in presence of Nafion and without Nafion. Both groups of samples (with and without Nafion), exhibit similar trend with higher negative zeta potential when ILs are introduced to the system because of their higher ionic strength, specifically in the case of Pt/C-($[C_2mim]^+[NTf_2]^-$) and Pt/C-($[C_4dmim]^+[NTf_2]^-$). Generally speaking, a zeta potential smaller than the agglomeration threshold, $|\xi|$<15mV, indicates the instability of the nanoparticles, which was only observed for pristine Pt/C suspension[36] without Nafion. Impregnating the carbon support with IL, significantly increased the absolute value of zeta potential. This indicates that ILs stabilized the Pt/C suspension and prevented it from agglomeration. Because ILs are charged, the electrostatic repulsion between Pt/C with ILs will be stronger than for pristine Pt/C[37]. Water uptake and zeta potential for Pt/C with ILs scale linearly. Pt/C catalyst layer has the least water uptake and least absolute value of zeta potential (−11 mV), whereas Pt/C-($[C_4dmim]^+[NTf_2]^-$) has the largest water uptake and the largest absolute value of zeta potential (−29 mV). When Nafion was added, zeta potential values for all the samples increased in absolute value to the range of −32 to −44 mV. Since the concentration of Nafion was high, I/C of 0.8, the electrostatic interactions between the Nafion chains can also help promote aggregate structure and higher local surface charge density[38].

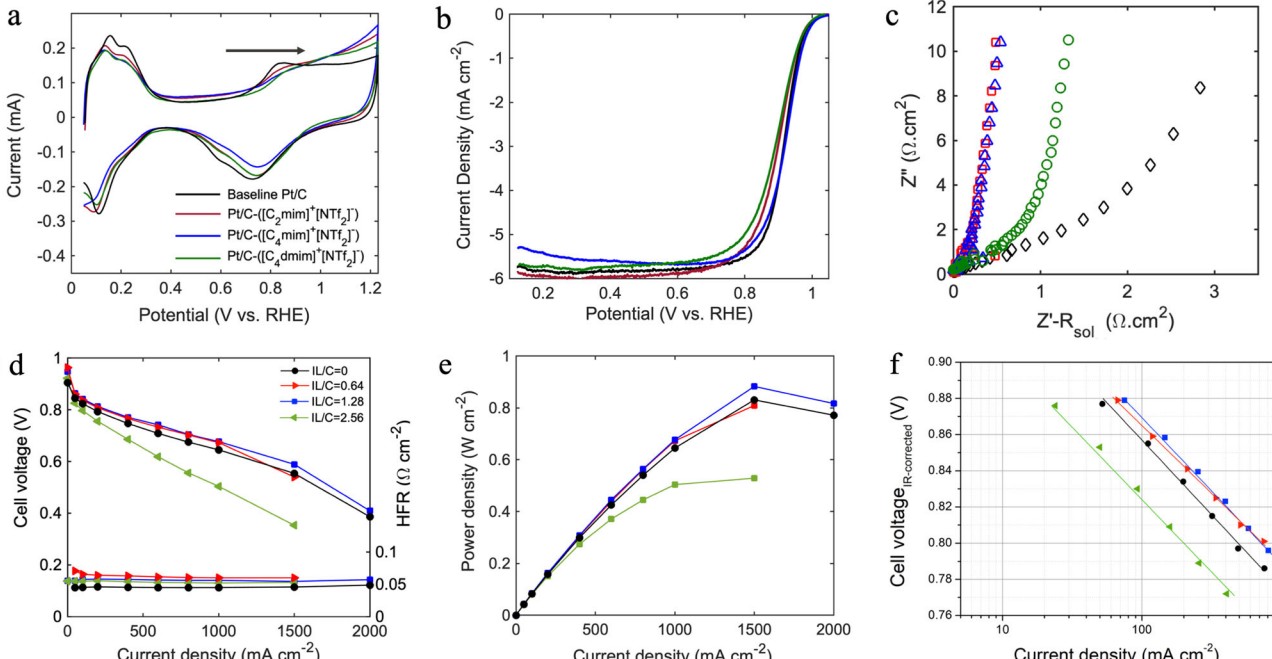

**Fig. 3 | Electrochemical characterization of baseline Pt/C vs. IL-modified Pt/C samples in rotating disk electrode setup and IL/C ratio optimization in MEAs containing various loading of 1-butyl-3-methylimidazolium bis(tri-fluoromethylsulfonyl)imide ([C₄mim]⁺[NTf₂]⁻) evaluated at 80 °C and 100% RH. a** Cyclic voltammograms of baseline Pt/C and IL-modified Pt/C in N₂-saturated 0.1 M HClO₄, **b** polarization activity in O₂-saturated 0.1 M HClO₄ at a rotation speed of 1600 rpm. **c** Nyquist plots of impedance spectra under N₂-saturated 0.1 M HClO₄ for Pt/C-Nafion and Pt/C-IL-Nafion catalyst layers. **d** H₂/air polarization curves of MEAs with IL/C ratio of 0.64, 1.28. 2.56 vs. baseline Pt/C in fuel cell, **e** power density obtained at 150 kPa$_{abs}$ total pressure, **f** H₂/O₂ Tafel plots measured at 150 kPa$_{abs}$ total pressure in 5 cm² differential cells, cathode: HSA Pt/C or HSA Pt/C-IL with loading 0.13 ± 0.02 mg cm⁻², anode: LSA Pt/C with loading 0.1 mg cm⁻².

Consequently, larger negative zeta potential was observed for Nafion-containing inks.

## RDE prescreening and IL/C ratio optimization in MEA

Baseline and IL-modified Pt/C inks were prepared with optimized amount of IPA and water, Nafion/C and IL/C ratio of 0.3 and 2.56, respectively. Catalyst films were then deposited with the procedure described Methods section having final Pt loading of 40 μg/cm²$_{geo}$. The electrochemical properties of all the catalyst layer films are provided in Table S3. Cyclic voltammograms of baseline and IL-modified Pt/C samples in Fig. 3a showed no extra faradaic processes other than those generally happening on Pt/C, with H$_{UPD}$ peak observed between 0 and 0.4 V vs. RHE, and surface-oxygenated species (OH$_{ad}$ and O$_{ad}$) beyond 0.6 V vs. RHE. By the addition of all three ILs to the Pt/C catalyst, the ECSA decreased because of blocking effect of ILs by their selective occupation of the Pt active sites, competition adsorption of their ions with hydrogen, and suppression of Pt-H bonding strength due to the direct ligand effect[8,29,39,40]. While observed ECSA for baseline Pt/C was 91 m² g⁻¹, ECSA for Pt/C-([C₂mim]⁺[NTf₂]⁻), Pt/C-([C₄mim]⁺[NTf₂]⁻), and Pt/C-([C₄dmim]⁺[NTf₂]⁻) decreased to 88.7, 76.54, and 73.7 m² g⁻¹, respectively. The onset potential of the Pt-oxide formation was also shifted to higher potentials and the area under OH$_{ad}$ and O$_{ad}$ peak decreased indicating that the formation of oxygenated species was suppressed resulting in lower coverage of Pt surface. IL molecules potentially protected the Pt active sites from being occupied by non-reacting oxygenated species; thus, facilitating the ORR kinetics as strong Pt-O bonding is known to slow ORR[41]. The structure-property relationship of these three ILs originates from both their interactions with Pt surface and induced hydrophobic microenvironment[27]. Elongation of alkyl chain can result in stronger interactions with Pt terrace sites. As shown in CV voltammograms and Table S3, by elongation of side chain from C₂ in Pt/C-([C₂mim]⁺[NTf₂]⁻) to C₄ in Pt/C-([C₄mim]⁺[NTf₂]⁻), the ECSA decreased indicating that butyl chains have blocked some Pt active sites. However, side chain interactions with those active sites

make them less prone to formation of oxygenated species[27]. Pt/C-([C₄dmim]⁺[NTf₂]⁻) has an additional interactive methyl group and showed super hydrophobic properties compared to the other two ILs; therefore, it provides the most dominant suppression of Pt-oxide formation. [C₄dmim]⁺[NTf₂]⁻ is the most effective IL in protecting Pt active sites from being oxidized by repelling water molecules and [C₂mim]⁺[NTf₂]⁻ has the least ECSA drop and passivation of Pt active sites. However, from polarization curves displayed in Fig. 3b it can be noted that only Pt/C-([C₄mim]⁺[NTf₂]⁻) catalyst film demonstrated a potential improvement in ORR kinetics as the half-wave potential (E$_{1/2}$) is slightly shifted toward higher potential compared to the baseline.

There have been extensive investigations in the literature to show ORR improvement in IL-modified Pt/C catalysts using RDE[24]. Huang et al.[29] indicated MA enhancement factor of about 1.2 for both [MTBD][beti] and [MTBD][C₄F₉SO₃]-modified HSA Pt/C. These samples improved SA by a factor of 1.4 and 1.7. The degree of performance enhancement is in a great agreement with our observations in which the MA and SA enhancement factors for HSA Pt/C-([C₄mim]⁺[NTf₂]⁻) were 1.2 and 1.7, respectively. Zhang et al.[27] explored the ORR enhancement on low surface area Pt/C using imidazolium based ILs and obtained MA enhancement factor of about 3 for their best performing IL that has similar chemical structure to ([C₄mim]⁺[NTf₂]⁻) and has the same cationic alkyl chain length. Yan et al.[30] also reported up to three-fold enhancement in MA and SA of Pt/C catalyst by replacing Nafion with a poly(ionic liquid) in which the IL segment acts as a proton conductor. Figure 3c compares the Nyquist plots of baseline and IL-modified Pt/C catalyst layers. The linear portion of the Nyquist plot data at intermediate frequencies is used for the calculation of the effective catalyst layer protonic resistance. The fitted line's intercept on the x-axis is considered to be R$_{H⁺}$/3[42]. Baseline Pt/C (in presence of Nafion) showed a noticeably higher proton diffusion resistance of 3.27 Ω cm² when compared to the catalyst layers with the mixture of Nafion and IL. The values for R$_{H⁺}$ can be found in Table S3. Specifically, Pt/C-([C₂mim]⁺[NTf₂]⁻) and Pt/C-([C₄mim]⁺[NTf₂]⁻) expressed superior

proton conductivity making them a great candidate for investigating the role of proton conduction in improving ORR catalytic activity when nanoconfinements are induced by ILs.

Furthermore, it has been shown that ORR activity enhancement using ILs is also attributed to the interfacial hydrogen-bonding between IL's cation and adsorbed intermediate oxygenated species of ORR ($O_{ad}$ and $OH_{ad}$). In this study, N-H$^+$ bonds of the imidazolium cation form hydrogen-bonding with $OH_{ad}$; therefore, it enables faster proton tunneling from IL to Pt active sites through intermediate species (N-H$^+$ + $OH_{ad}$ + e$^-$ → $H_2O$ + N). Hydrogen-bonding structure can regulate the exchange current density of ORR and reaction rate constant based on the vibrational features of the intermediates[34].

As Pt/C-($[C_4mim]^+[NTf_2]^-$) with IL/C ratio of 2.56 expressed a potential in improving the catalytic activity in RDE level, we introduced various loadings of this IL to the catalyst layer and tested them in fuel cell hardware in order to optimize MEA-level performance. In order to prepare the ink for MEA fabrication, impregnated powders were redispersed in the Nafion solution in water and IPA. The ink suspension was then coated, dried, and hot-pressed on Nafion membrane for MEA manufacture as described in Methods MEA fabrication and assembly section. To investigate the Pt/C-ILs interface evolution during MEA preparation, we performed a complementary experiment. Details of this experiment and observations (Fig. S5) are summarized in the supplementary information. We believe depending on the IL chemical structure and pore filling degree, a portion of the impregnated IL in the catalyst powder would redissolve in ink's Nafion solution as IL is soluble in the mixture of water and IPA. Then, it deposits along with Nafion during coating and drying. One can assume that IL molecules on the surface, in the macropores and larger mesopores would dissolve faster in the Nafion solution compared to the ones packed in the micropores and smaller mesopores. Figure 3d, e represents the polarization curve and power density of the MEAs with various loadings of ($[C_4mim]^+[NTf_2]^-$). The electrochemical properties of the MEAs are summarized in Table S4. In order to reach the optimum performance, MEA with IL/C ratio of 2.56 had to undergo 8 voltage recovery cycles, where the cell was held at low potential and oversaturated condition (150% RH) to remove excess surface-blocking species including sulfates. The maximum current density obtained from this cell was only 1.5 A cm$^{-2}$ confirming there was a thick layer of IL causing 69% ECSA loss and a high mass transport resistance at high current densities. Although the SA at 0.9 V increased compared to the baseline, the MA was only 110 A g$_{Pt}^{-1}$. By cutting the amount of IL into half, IL/C ratio of 1.28, MEA showed enhanced performance across all current densities, with optimum peak power density of 0.909 W cm$^{-2}$. MA and SA obtained from the Tafel plot at 0.9 V in Fig. 3f increased to 347 A g$_{Pt}^{-1}$ and 697 µA cm$^2_{Pt}$, respectively. The amount of ($[C_4mim]^+[NTf_2]^-$) was then reduced to half, causing the MA and SA at 0.9 V drop to 173 A g$_{Pt}^{-1}$ and 509 µA cm$^2_{Pt}$. Although the ECSA loss in this case was only 16%, the amount of IL was insufficient to improve proton conductivity efficiently in the smaller mesopores, specifically at high current densities where the cell current was limited to a maximum of 1.5 A cm$^{-2}$ as a result. All three IL-containing MEAs showed similar local oxygen mass transport resistance within the range of 0.261 ± 0.006 s cm$^{-1}$ expressing that local mass transport resistance depends on the chemical nature of the IL rather than its loading (Table S4 and Fig. S4).

The reason we believe IL/C ratio of 1.28 is the optimum ratio for MEA fabrication is that based on our study summarized in the supplementary information, less than 30% of the IL is covering the surface or is in the bigger pores. Therefore, about 70% of the IL is deposited in the micro and smaller mesopores. This portion of IL would not dissolve as fast as the ones on the surface and would stay in the smaller pores facilitating proton transfer. One the other hand, at high IL loadings (IL/C = 2.56), there is a layer of IL covering the whole surface that would redeposit with Nafion in the catalyst layer causing high ionic resistance.

## In situ MEA characterization of baseline Pt/C vs. IL-modified catalysts

Figures 4 and 5 and Table 2 show the electrochemical characteristics of baseline Pt/C and all three IL-modified MEAs with optimized IL/C ratio of 1.28. Figure 4a displays the optimal polarization curves obtained after two and three voltage recovery cycles for baseline Pt/C and IL-modified MEAs, respectively. Pt/C-($[C_4mim]^+[NTf_2]^-$) showed a significant improvement across all current densities, while MEA containing ($[C_2mim]^+[NTf_2]^-$) showed similar polarization curve to the baseline with slightly enhanced potential in current densities between 400 and 1500 mA cm$^{-2}$. Neither Pt/C-($[C_2mim]^+[NTf_2]^-$) nor Pt/C-($[C_4dmim]^+[NTf_2]^-$) promoted the performance in kinetic and mass transport-limited regions. Additionally, all the IL-containing MEAs showed higher OCV than that of the baseline which can be attributed to the exposure of Pt to ILs' imidazolium ions, structural changes of the EDL, interfacial electron exchange between IL and the surface of carbon that led to positive shift in the OCV[24,43,44]. Furthermore, Pt/C-($[C_2mim]^+[NTf_2]^-$) and Pt/C-($[C_4mim]^+[NTf_2]^-$) MEAs reached peak power densities of 0.856 and 0.909 W cm$^{-2}$ at 61.2% and 63.9% voltage efficiencies, while baseline's peak power density was only 0.831 (Fig. 4b and Table 2). As displayed in Fig. 4c, cyclic voltammograms confirmed that the presence of all three ILs protect the Pt surface from formation of non-reactive oxygenated species by shifting the peak to higher potentials. ECSA loss for modified catalysts in MEA-level measurement was 26.8%, 31.1%, 48.3% for Pt/C-($[C_2mim]^+[NTf_2]^-$), Pt/C-($[C_4mim]^+[NTf_2]^-$), and Pt/C-($[C_4dmim]^+[NTf_2]^-$), respectively. As discussed before in the RDE section, this could be related to the molecular structure of the ILs and the interaction of the cationic alkyl chain with Pt terrace site. In correlation with previously reported observations[27], increase in the number of alkyl chains, as well as their length can increase the interaction with Pt sites. In addition, MEA-level measurements revealed much higher ECSA losses compared to RDE set-up. RDE experiments are carried out in aqueous environment in which all the Pt within the pores is accessible by protons through adsorbed water films. On the other hand, ECSA measurement in MEAs occur at 100% RH where IL molecules are expected to have different interactions with Pt compared to fully aqueous environment.

The EIS data fitting at high frequencies (Fig. S6) illustrates that Pt/C-($[C_2mim]^+[NTf_2]^-$) and Pt/C-($[C_4mim]^+[NTf_2]^-$) catalyst layers in the presence of Nafion have significantly enhanced proton conductivity as high as 3.95 and 2.34 S m$^{-1}$, while Pt/C-($[C_4dmim]^+[NTf_2]^-$) shows similar proton conduction to the baseline (Fig. 5d). Moreover, as it is summarized in Fig. 5c, d, modified MEAs with $[C_2mim]^+[NTf_2]^-$ have ca. ×2 higher local oxygen mass transport resistance than that of baseline, which is offset by the enhanced high ionic conductivity (also ca. ×2) resulting in a similar polarization curve to the baseline except in the mass transport-limited region where oxygen mass transport resistance become more critical. The similar trend is also observed for Pt/C-($[C_4dmim]^+[NTf_2]^-$), where the lower ionic conductivity compared to the baseline is offset by the lower local oxygen mass transport resistance due to its lower filling degree. However, in this case, the notable loss in the electrochemical active area is responsible for the significant drop in the catalytic activity as the MA obtained at 0.9 V decreased to 121 A g$_{Pt}^{-1}$ (Table 2 and Fig. 5a). Although ($[C_4dmim]^+[NTf_2]^-$) expressed the lowest transport resistance compared to all other samples and is more capable of suppressing Pt-oxide formation, the interplay of significant ECSA loss, lower ionic conductivity and filling degree, makes it insufficient in promoting catalyst ORR activity. Local oxygen mass transport resistance for all four MEAs summarized in Fig. 5c was obtained from y-intercept of mass transport resistance line as a function of total pressure presented in Fig. S7. Mass and specific activities at 0.9 V summarized in Fig. 5a, b were also calculated from Tafel plots presented in Fig. 4d.

In contrast, Pt/C-($[C_4mim]^+[NTf_2]^-$) showed superior electrochemical properties that highlights the importance of how optimized

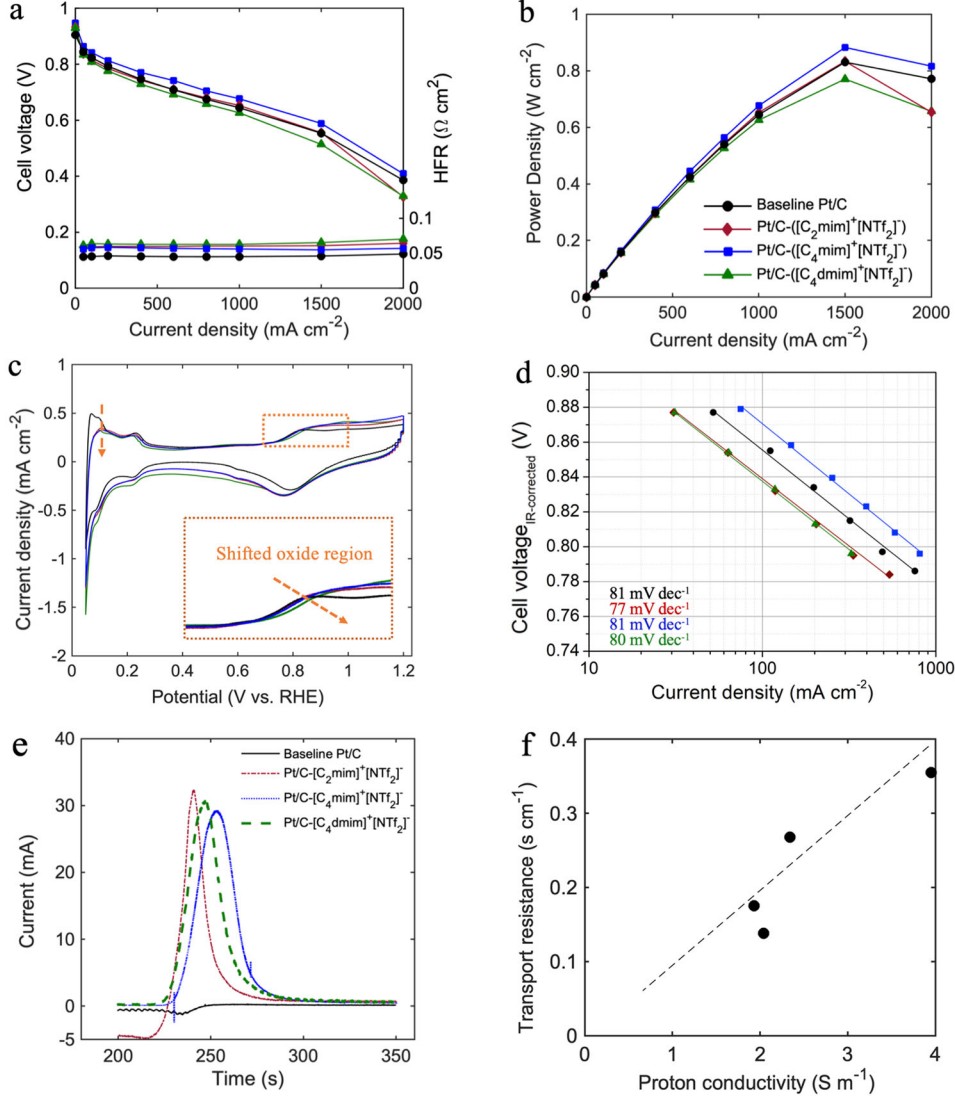

**Fig. 4 | Electrochemical characterization of IL-containing MEAs obtained after three voltage recovery cycles vs. baseline Pt/C after two voltage recovery cycles evaluated at 80 °C and 100% RH. a** $H_2$/air polarization curves, **b** power density at 150 kPa$_{abs}$ total pressure, **c** cyclic voltammograms recorded in $H_2/N_2$ and 100 kPa$_{abs}$ total pressure, **d** $H_2/O_2$ Tafel plots at 150 kPa$_{abs}$ total pressure, **e** CO-displacement at 0.35 V at 100 kPa$_{abs}$ and 40 °C, **f** correlation between local oxygen mass transport resistance and proton conductivity in 5 cm$^2$ differential cells, cathode: HSA Pt/C or HSA Pt/C-IL with loading $0.13 \pm 0.02$ mg cm$^{-2}$, anode: LSA Pt/C with loading 0.1 mg cm$^{-2}$.

molecular structure and induced hydrophobic nanoconfinement can alter the activity of the catalyst toward ORR. As illustrated in Fig. 5a, b and Table 2, Pt/C-([C$_4$mim]$^+$[NTf$_2$]$^-$) showed enhanced mass activity of ~350 A g$_{Pt}^{-1}$ and specific activity of 700 µA cm$^2_{Pt}$ which are 20% and 75% higher than that of the baseline, respectively. As previously shown in this work, ([C$_4$mim]$^+$[NTf$_2$]$^-$) is capable of filling the pores sufficiently and its optimal hydrophobicity is effective enough to protect the Pt active sites from Pt-oxides while the ECSA loss is only 31%. On the other hand, it can form a well-connected proton pathway with Nafion as the zeta potential measurement showed there is a small repulsion between Nafion chains and IL molecules. From the chemical structure point of view, enhancement in the ionic conductivity can also be attributed to an interplay of various mechanisms: (1) [NTf$_2$]$^-$ serving as proton defect (hopping site) and carrier, (2) proton migration through Grotthuss mechanism as the conformational flexibility of [NTf$_2$]$^-$, distributed negative charge, parallel $\pi - \pi$ stacking in the structure of imidazolium facilitate the hydrogen-bonding formation between H atoms of the imidazolium ring and N and O atoms of the [NTf$_2$]$^-$. Furthermore, since ([C$_4$mim]$^+$[NTf$_2$]$^-$) induces a minimal hydrophobic microenvironment,

it can still retain some extent of water near Pt active sites promoting the proton conduction via both Grotthuss and vehicular mechanisms in form of $H_3O^+$ [31,45–48] Figure S8 compares the polarization curves of Pt/C-([C$_4$mim]$^+$[NTf$_2$]$^-$) in wet (100% RH) and dry conditions (75%, 50%, 30% RH) vs. the baseline Pt/C. Earlier study showed dramatic loss of performance for MEA with IL under dry conditions[28]. Here, we see overall good performance for IL-filled Pt/C but not as high compared to the baseline at dry conditions. At dry conditions for the baseline catalyst, most of the Pt in contact with ionomer will be active, but Pt inside the mesopores will not; unless they are specifically engineered to condense water at these low RHs (30 and 50% RH). Since ionomer orientation near Pt with IL-filled layers is different, local hydrophobic environments that ILs induce are not favorable for performance in dry conditions. This is probably due to the fact that ILs' hydrophobic domains repel $SO_3^-$ groups and water molecules from IL|ionomer interface causing higher resistance at the interface. Therefore, catalyst layers have to be further reengineered if dry operation is the target.

Another important observation shown by Fig. 4e is the CO-displacement plots for the baseline and IL-modified Pt/C. Baseline

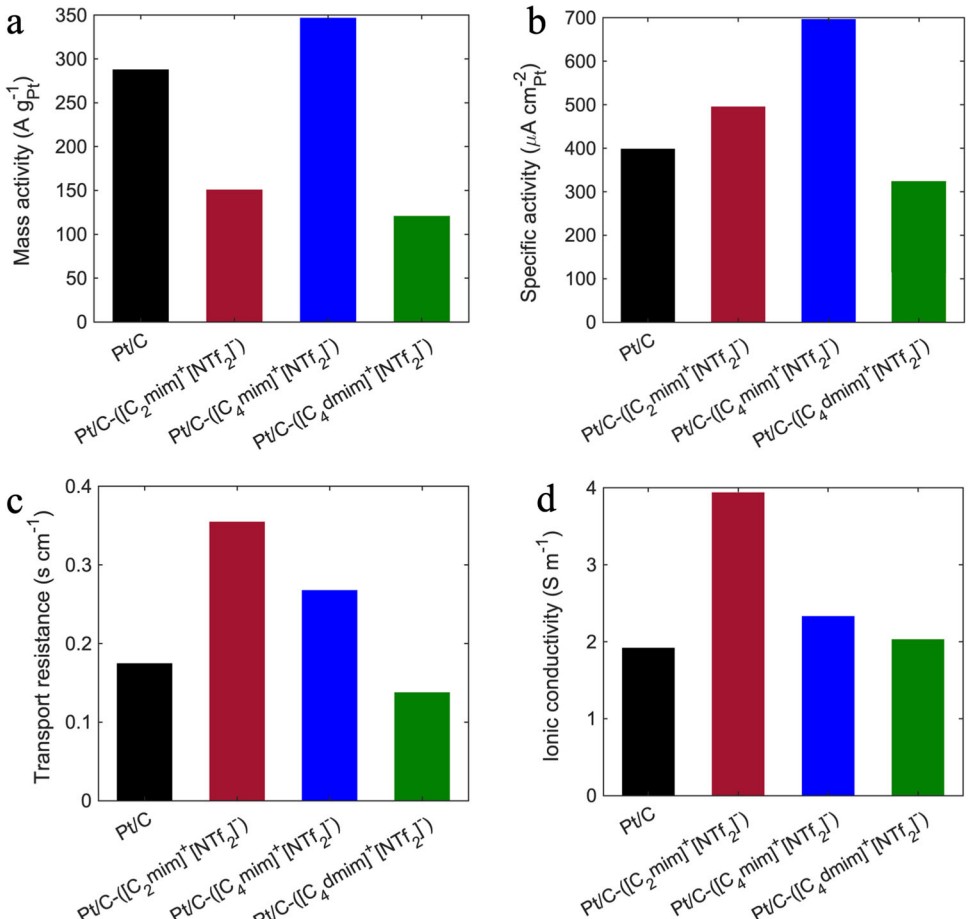

**Fig. 5 | Electrochemical characterization of IL-containing MEAs obtained after three voltage recovery cycles vs. baseline Pt/C obtained after two voltage recovery cycles evaluated at 80 °C and 100% RH. a** Mass and **b** specific activities at 0.9 V, 150 kPa$_{abs}$ and 100% RH, and **c** local oxygen transport resistance measured at 75% RH, and **d** ionic conductivity at 100% RH in 5 cm$^2$ differential cells, cathode: HSA Pt/C or HSA Pt/C-IL with loading 0.13 ± 0.02 mg cm$^{-2}$, anode: LSA Pt/C with loading 0.1 mg cm$^{-2}$.

**Table 2 | Electrochemical properties of baseline Pt/C MEA obtained after two voltage recovery cycles and Pt/C-IL MEAs obtained after three voltage recovery cycles in fuel cell set-up**

|  | Baseline Pt/C | Pt/C-([C$_2$mim]$^+$[NTf$_2$]$^-$) | Pt/C-([C$_4$mim]$^+$[NTf$_2$]$^-$) | Pt/C-([C$_4$dmim]$^+$[NTf$_2$]$^-$) |
|---|---|---|---|---|
| OCV (V) | 0.905 | 0.933 | 0.948 | 0.929 |
| Cell potential @ 0.8 A cm$^{-2}$ | 0.675 | 0.680 | 0.705 | 0.658 |
| Peak power density (W cm$^{-2}$) | 0.831 | 0.856 | 0.909 | 0.804 |
| Mass activity at 0.9 V (A g$^{-1}$$_{Pt}$) | 288 | 151 | 347 | 121 |
| Specific activity at 0.9 V (µA cm$^2$$_{Pt}$) | 399 | 496 | 697 | 324 |
| ECSA (m$^2$ g$^{-1}$) | 72.1 | 52.8 | 49.7 | 37.3 |
| Proton conductivity (S m$^{-1}$) | 1.93 | 3.95 | 2.34 | 2.04 |
| Local O$_2$ mass transport resistance (s cm$^{-1}$) | 0.175 | 0.355 | 0.268 | 0.138 |

catalyst layer shows a reduction peak when CO is introduced at 0.35 V indicating that CO is displacing SO$_3^-$ groups, as Pt is positively charged at this potential and SO$_3^-$ groups adsorb on a positive surface. On the other hand, oxidation peak is observed for all three catalyst layers with IL, indicating that CO is displacing a positive charge from the surface of Pt. Interestingly, XPS data summarized in Fig. S3 and Table S2 showed no substantial chemical changes in Pt and Pt-oxide states after the ex situ XPS CO dosing confirming that IL molecules had no adverse interactions with Pt surface. These ex situ experiments were not performed under applied potentials and most likely Pt does not strongly interact with the imidazolium groups at OCV. However, under applied potentials in the in situ environment this was not the case. The

oxidative peak observed during IL-induced displacement of CO on Pt is attributed to the imidazolium groups adsorbing on surface. Sabota et al.[39] previously have shown that imidazolium cations have electronic interactions with Pt through π-system of the imidazolium and the formation of carbene species at the surface. This observation can be further confirmed with our CO stripping data shown in Fig. S9. The negative shift of CO stripping peak to lower potentials indicates a weaker Pt-CO$_{ads}$ bonding in presence of ILs due to the ligand-like interactions. Therefore, imidazolium-derived ILs significantly modify Pt d-band centers via ligand effects. During the CO displacement in presence of ILs, IL molecules preferentially interact with Pt(111) terrace sites, while CO selectively adsorbs on the edges. It can be seen in

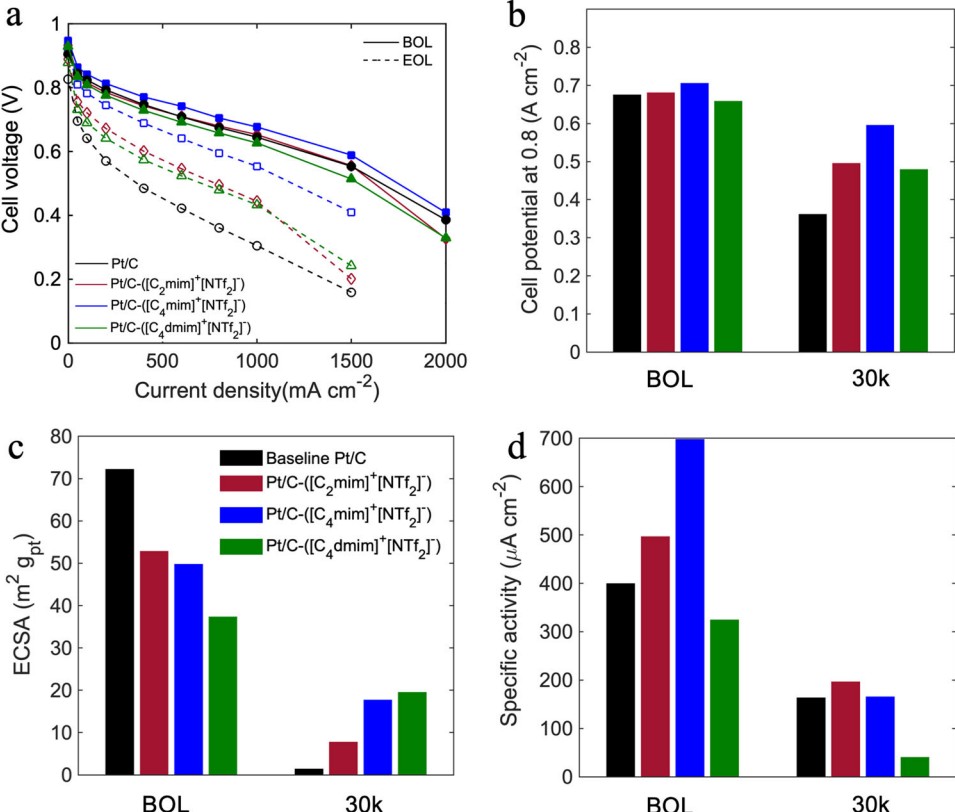

**Fig. 6 | Comparison of electrochemical properties at the beginning of life (BOL) and after 30,000 cycles (EOL) for Pt/C baseline vs. Pt/C-([C₂mim]⁺[NTf₂]⁻), Pt/C-([C₄mim]⁺[NTf₂]⁻), and Pt/C-([C₄dmim]⁺[NTf₂]⁻) MEAs. a** Comparison of $H_2$/air polarization curves, **b** cell potential at 0.8 A cm⁻², **c** mass activities obtained at 0.9 V vs. RHE, and **d** ECSA for all MEAs at BOL vs. EOL (after 30k cycles). BOL electrochemical properties were obtained after two and three voltage recovery cycles for baseline and IL-containing MEAs, respectively. EOL properties were obtained after one voltage recovery cycle for all MEAs.

Fig. S9 that ([C₄mim]⁺[NTf₂]⁻) shows the smallest left shift compared to the other two ILs. One can comprehend that ([C₄mim]⁺[NTf₂]⁻) can induce the strongest bonding between Pt and reactant molecules ($O_2$ and CO) compared to the other two IL[9]; while it effectively prevent $SO_3^-$ groups adsorption onto the surface of Pt, adding to its catalytic activity. Figure 4f reveals that there is a tradeoff between the proton conductivity and local oxygen mass transport resistance as a higher proton conductivity comes at the expense of larger oxygen transport resistance. Therefore, based on the MEA design needs, an optimal point should be selected which was the case for MEA containing ([C₄mim]⁺[NTf₂]⁻) in this work.

## Durability studies

In order to probe the degradation behavior of the modified catalysts, DOE-specified catalyst AST was conducted that involved load cycling in range of 0.6–0.95 V each for 3 s and 30,000 cycles. Although HSA Ketjenblack Pt/C catalysts are generally electrochemically active due to the lower ionomer poisoning, they tend to lose their catalytic activity dramatically at 800 mA cm⁻² after only 30 h of accelerated testing[49]. Figure 6 and Table S5 show the evolution of electrochemical properties for IL-containing MEAs vs. baseline at the beginning of life (BOL) and end of life (EOL). All four MEAs showed performance loss over the course of 30,000 AST cycles which took about 50 h. The comparison of polarization curves at BOL and EOL (Figs. 6a and S10) reveals that all the ILs have significantly improved the durability of the cells. All the IL-containing MEAs showed similar trend at the BOL and EOL, while baseline exhibits more severe loss in medium and high current density area. As demonstrated in Fig. 6b, baseline's potential at 0.8 A cm⁻² dropped from 0.675 to 0.361 V (reduced by 0.314 V); however, the potential decrease for Pt/C-([C₄mim]⁺[NTf₂]⁻) was only 0.11 V.

Pt/C-([C₂mim]⁺[NTf₂]⁻) and Pt/C-([C₄dmim]⁺[NTf₂]⁻) both exhibited a potential decrease of 0.18 V. Figure 6c represents the ECSA of all the MEAs at BOL and EOL. As can be seen in Fig. 6c, baseline experienced the most significant ECSA loss from 72.1 to 1.32 m² g⁻¹ and retained only 2% of the initial ECSA. On the other hand, Pt/C-([C₄mim]⁺[NTf₂]⁻) and Pt/C-([C₄dmim]⁺[NTf₂]⁻) maintained 36%, and 48% of the ECSA, and exhibited surface area of 17.63 and 19.45 m² g⁻¹ even after 30,000 cycles, respectively. Pt/C-([C₂mim]⁺[NTf₂]⁻) ECSA decreased from 52.8 to 7.60 m² g⁻¹ maintaining 15% of the surface area. We can conclude that bigger IL molecules with longer alkyl chains and more complex chemical structure have more interactions with Pt and the support and are harder to be washed out of the pores; therefore, Pt in contact with ILs will be protected from coarsening. We previously have shown the same effect in HSA Pt/C in which Pt in contact with water degraded more compared to Pt in contact with ionomer[17]. Since most of the Pt nanoparticles are dispersed within the pores and are not in contact with ionomer, ILs' role become more dominant in preventing Pt degradation. In confined environment, Pt dissolution rate in IL-filled mesopores is lower compared to water-filled pores, as ILs prevent Pt ions from migration in similar way ionomer does. It is also possible that the amount of PtOx is lower as shown by CVs, and PtOx dissolution is higher. Therefore, by mitigating oxide growth on Pt surface, ILs also prevent dissolution. ILs can limit the migration of Pt nanoparticles and further collision and coalescence. Figure 6d displays the trends in specific activities obtained at 0.9 V during the AST cycling. Both baseline and Pt/C-([C₂mim]⁺[NTf₂]⁻) lost 60% of the specific activity, but Pt/C-([C₂mim]⁺[NTf₂]⁻) still exhibits higher SA of 196 μA cm⁻² than that of baseline. Pt/C-([C₄mim]⁺[NTf₂]⁻) showed comparable specific activity to baseline although it showed higher roughness factor. These results confirm that catalysts containing ILs are more active toward

ORR after 30,000 cycles due to (1) higher ECSA retention and (2) sufficient proton conductivity of remaining IL molecules within the pores after the degradation of Nafion. Figure S11 represents the changes in peak power density for MEAs. While the peak power efficiency at 30,000 cycle lifetime for the baseline was only 42%, it increased to 55%, 71%, and 57% for Pt/C-($[C_2mim]^+[NTf_2]^-$), Pt/C-($[C_4mim]^+[NTf_2]^-$), and Pt/C-($[C_4dmim]^+[NTf_2]^-$), respectively.

In summary, we identified the design criteria for modification of catalyst layers by incorporation of midazolium-derived ILs. These criteria included high $\Delta pK_a$ value for sufficient proton transfer, high oxygen solubility, superior thermal and electrochemical stability, etc. Water uptake and zeta potential measurements brought an insight into the hydrophobicity degree of these ILs that dictates the conformational structure of the Nafion chains as they reorient near Pt surface and within the pores. In addition, the study of physical characterization and EIS fitting demonstrated that proton conductivity is proportional to pore filling degree which are the highest for Pt/C-($[C_2mim]^+[NTf_2]^-$). On the other hand, in situ electrochemical measurements showed that higher proton conductivity comes at the expense of larger local oxygen transport resistance. Therefore, the best performing IL should be the one that achieved a tradeoff between these two. Moreover, based on the correlation between in situ fuel cell testing and ex situ physical characterization, we established a clear understanding of ILs role in altering the interactions between Pt, IL, and the ionomer at triple-phase interface. We concluded that a hydrophobic IL with a minimal water uptake induces the least electrostatic repulsion on Nafion chains, while keeping the water layers near Pt surface that promotes the proton conductivity. More importantly, our CO displacement studies showed the predominant role of ILs in protecting Pt surface from $SO_3^-$ poisoning. Our ($[C_4mim]^+[NTf_2]^-$)-modified Pt/C catalyst with optimal hydrophobicity, proton conductivity, and oxygen transport resistance achieved a peak power density of 0.909 W cm$^{-2}$ and enhanced MA and SA by 20% and 75%, respectively. Under the DOE catalyst AST testing protocol, Pt/C-($[C_4mim]^+[NTf_2]^-$) achieved an improved durability compared to the baseline as the ILs principally protect Pt from coarsening. We believe that the design approach and concepts presented in this paper can provide a promising way to future design of highly durable and active Pt-based catalysts for PEFC application.

## Methods
### Materials
The Pt/C electrocatalyst was 40 wt% platinum on high surface area Ketjenblack obtained Fuel Cell Store (College station, Texas, United States). HClO$_4$ (70%), 2-Propanol BioReagent (99.5%) were purchased from Millipore Sigma (Burlington, Massachusetts, United States). Imidazolium-derived ILs including 1-ethyl-3-methylimidazolium bis(trifluoromethylsulfonyl)imide ($[C_2mim]^+[NTf_2]^-$), 1-butyl-3-methylimidazolium bis(trifluoromethylsulfonyl)imide ($[C_4mim]^+[NTf_2]^-$), and 1-butyl-2,3-dimethylimidazolium bis(trifluoromethylsulfonyl)imide ($[C_4dmim]^+[NTf_2]^-$) were obtained from Solvionic Co., (Toulouse, France). All the reagents were used as received without further purification.

### Pt/C modification with ILs
The ILs impregnation method into the catalyst powder was adopted from ref. 29. In total, 9.5 mg of the Pt/C electrocatalyst was first placed in a vial and wet by 0.5 ml of DI water. Based on the desired ratio of IL/C, IL solution with the concentration of 5 mg/ml was prepared. A solution of the imidazolium-derived ILs in IPA was prepared with 5 mg ml$^{-1}$ concentration. Then, the IL solution was mixed with the wetted Pt/C electrocatalyst, and ultrasonicated for 20 min. The mixture was then stored on a magnetic hotplate stirrer at 45 °C and the solvents were evaporated slowly into the ambient atmosphere while continuously mixed with a magnetic stir bar. The obtained powder was further dried under vacuum at room temperature overnight. This impregnation method was used to integrate imidazolium-derived ILs into Pt/C catalyst with different IL/C ratios of 0.64, 1.28, and 2.56.

### Rotating disk electrode (RDE) testing
**Ink preparation and deposition on RDE electrode.** In order to achieve final Pt loading of 40 µg$_{Pt}$ cm$^{-2}$$_{geo}$ on the disk, 5 mg of the impregnated catalyst was mixed with 996.5 µl of MilliQ H$_2$O:IPA (24 wt% IPA), and 8.12 µl of a 5 wt% Nafion solution. To promote the uniform distribution of the catalyst, the ink was ultrasonicated for 1.5 h. The ultrasonic bath temperature was maintained below 40 °C by adding ice to preserve Nafion structure[50]. A 10 µl aliquot of the catalyst ink was pipetted in two parts (5 µl each) on glassy carbon tip mounted on an inverted rotator shaft while rotating at 60 rpm. Then, the rotation gradually increased to 500 rpm, and the ink dried under ambient conditions while rotating.

**Electrochemical cell preparation.** The electrochemical cell glassware is custom-made for the purposes of these experiments by Adams & Chittenden (Berkeley, California, United states). References electrode is connected to the main body of the cell by a salt bridge, to avoid any contamination[50]. To obtain reproducible measurements and absolute values of activities, glassware must be cleaned thoroughly of organic impurities; therefore, before each experiment, the electrochemical cell and the glassware were soaked in piranha solution (solution of concentrated H$_2$O$_2$ and H$_2$SO$_4$ with equivalent volumes) overnight. Then, they were all washed thoroughly with MilliQ water (18.2 MΩ) and boiled for 2 min to remove adsorbed H$_2$SO$_4$ from Piranha. To obtain a uniform catalyst layer, the glassy carbon electrode was polished using alumina slurry (5 µm, 0.3 µm and then 0.05 µm), followed by excessive cleaning with acetone, ethanol, and MilliQ water. All the electrochemical measurements were performed by a Gamry potentiostat (1010E interface), using the surface mode. The electrolyte was 0.1 M HClO$_4$ prepared with MilliQ water (18.2 MΩ). A Hydrogen reference electrode (Hydroflex, Gaskatel, Kassel, Germany) and a Pt wire were used as reference electrode and counter electrode, respectively. All potentials in this work are reported against a reversible hydrogen electrode (RHE).

**Electrochemical characterization.** The electrochemical characterization was performed as follows. The cell was first pre-conditioned by purging with N$_2$ for 20 min. To clean the surface of the electrocatalyst, activation cyclic voltammetry (CV) was performed by sweeping the potential between 0.05 V and 1.23 V vs. RHE at 500 mV s$^{-1}$ for 100 cycles. Then, five stable CVs were recorded in the same potential range at 20 mV s$^{-1}$. EIS was performed at 0.40 V vs. RHE, from 2 MHz to 1 Hz to measure ohmic resistance and protonic resistance (R$_H^+$). A linear sweep voltammetry (LSV) was also recorded under N$_2$-saturated condition at 1600 rpm between 0.05 and 1.05 V vs. RHE at 20 mV s$^{-1}$ for background correction. Then, the cell was purged with O$_2$ for 20 min. ORR activity was evaluated by LSV at 5 mV s$^{-1}$ toward an anodic scan (0.1 V–1.05 V vs. RHE) at 1600 rpm.

**MEA fabrication and assembly.** Modified and unmodified Pt/C catalyst were weighed and dispersed in a mixture of DI water and 1-propanol resulting in a final concentration of 0.003 g/ml. Optimum cathode inks were moderate-water content ink with 51 wt% water in solvent. The D521 Nafion™ Dispersion—Alcohol based 1100 EW at 5 wt% ionomer was then added to the ink solution to get ionomer to carbon I/C ratio of 0.8. Then, the ink was ball milled along with ZrO$_2$ balls for 18 h and wet film applicator with 200 µm thickness was used to coat on fiber reinforced Teflon substrate (250 µm thick) using tape casting machine (13 mm s$^{-1}$). Coated decal was hot pressed on Nafion XL membrane (27.5 µm) at 155 °C for 3 min at ~0.1 kN cm$^{-2}$ of force. Weight of decal was measured before and after hot pressing to calculate the final Pt loading. The composition of MEAs and Pt loading were then

confirmed with X-ray fluorescence spectroscopy to be in a range of $0.13 \pm 0.02$ mg cm$^{-2}$. Catalyst coated membranes (CCMs) were then fitted into 5 cm$^2$ active area (1.35 cm × 3.7 cm) by impermeable PET subgaskets (8um thickness) to precisely span 14 parallel channels over 3.7 cm of DOE flow-field for differential condition as PET gaskets inhibited cross-channel convection. The CCMs were sandwiched between two Freudenberg H23C6 GDLs at 20% compression. The CCMs, GDLs, and polytetrafluoroethylene gaskets (27.5 μm PTFE and 120 μm fiber reinforced PTFE) were then placed between the flow fields, and the bolts were tightened at a torque of 13.5 N m.

**In situ fuel cell characterization.** Scribner 850e fuel cell test stand with maximum current load of 50 A from Scribner Associates, Connecticut, USA was used for performing AST and measuring performance metrics such as polarization curve, mass activity, oxygen transport resistance. AST was performed in H$_2$/N$_2$ environment at 80 °C and 100% relative humidity (RH) using a square-wave potential profile from 0.6 to 0.95 V with 3 s hold at each potential. VSP-BioLogic 4 A potentiostat was used for recording CV, LSV, and EIS.

The cell was first heated up to 60 °C and 100% RH. MEAs were conditioned by potential holds of 30 s at 0.8, 0.6, and 0.3 V until constant current was achieved. It usually took about 200 cycles to reach this point. Voltage recovery was performed at 0.2 V under H$_2$/air environment at 40 °C and 150% RH for 2 h. Polarization curves preceded with a voltage recovery step to retrieve all the recoverable losses. I–V curves generated under differential condition by holding the cell at constant currents for 3 min at 80 °C and 150 kPa total pressure with 100% RH. Polarization curves were stepped from high current densities to low current densities. The corresponding potentials at multiple points were measured and averaged over 3 min. Oxygen mass transport resistance was measured at 80 °C and 75% RH, with inlet dry mole fraction of O$_2$ in N$_2$ maintained at 1, 2, and 4%. The outlet pressure was controlled at 100, 125, 150, 200, kPa$_{abs}$. The limiting current was measured at constant voltages of 0.30, 0.24, 0.18, 0.12, and 0.06 V and held for 3 min[51]. The transient current response (I–t) resulting from the introduction of CO to an equilibrated electrode held at 0.35 V was measured as the adsorbed anions and cations will be displaced with CO through oxidation and reduction processes. Displacement charge will show the amount of charged surface species at 0.35 V. EIS was performed using 0.2 V DC voltage with AC amplitude being 5% of that of DC voltage under fully humidified H$_2$/N$_2$ environment, at temperature of 80 °C. A frequency range from 10,000 Hz to 0.1 Hz was used for these measurements. EIS data were then fit to obtain catalyst layer ionic conductivity values. A transmission line model was used for fitting the EIS spectra[52].

### Ex situ characterization

**Experimental parameters for N$_2$ physisorption.** Nitrogen gas physisorption was conducted with Micromeritics 3Flex Adsorption Analyzer to study the porosity and hydrophobicity of the Pt/C and Pt/C-IL samples. The samples were first degassed at 150 °C for 12 h before analysis. Nitrogen physisorption was performed at 77 K between a relative pressure (P/P$_0$) of 0 and 0.995, wherein a low pressuring dosing (2 cm$^3$ g$^{-1}$ STP) mode was used below 0.01 (P/P$_0$) for the microporous structure. The specific surface area was estimated by Brunauer–Emmett–Teller Model, and t-plot model was used to estimate the micropore volume/area and mesopore surface areas. The PSD above 2 nm was calculated with BJH model by using the adsorption isotherm. The PSD below 2 nm was estimated by non-linear DFT model assuming a slit pore geometry.

**Transmission electron microscopy (TEM).** Aberration corrected-TEM was performed on a JEOL JEM-ARM300CF microscope operated at 80 keV to minimize the beam damage to the sample. TEM samples were prepared by dispersing powders in the mixture of Millipore water

and IPA through ultrasonication. A drop less than 5 μl was then placed onto holey carbon-coated copper grids and further dried before placing into the TEM.

**Water uptake and contact angle measurements.** Samples for this measurement were also prepared using the same recipe explained earlier. The catalyst inks were coated evenly on both sides of an Ethylene Tetrafluoroethylene substrate. After having the ink coated on the substrate, the film was then cut into a 3 × 2 cm rectangle and was subjected to external contact angle measurement in order to evaluate the wetting properties using a Krüss K100 force tensiometer. The technique by which these experiments were performed is developed by Abbou et al.[53]. Each sample was suspended from the top of the instrument microbalance and the glassware vessel containing DI water was placed on top of the motorized platform. The sample was then immersed into the DI water for 1 cm at a constant rate of 10 cm min$^{-1}$. After having been submerged for 900 s in water, the vessel was removed causing a sudden increase in the mass detected by the microbalance. Using Wilhelmy Eq. (1) and the changes in the mass of each sample when it was removed from the water at 900 s, the external contact angle was calculated. $\Delta m$ is the changes in the mass of the catalyst layer after being removed from the water, $\rho_l$ is water's density, $V_i$ is the fraction of the sample immersed into the water and can be calculated by $V_i = Led$.

$$\theta = \cos^{-1}\left(\frac{-\Delta m.g + \rho_l g V_i}{2(L+e)\gamma}\right) \tag{1}$$

Details of the exact experimental details can be found in ref. 54. The amount of water uptake shown in Table 1 is calculated based on the difference between the initial weight of the sample and its weight after water absorption at 900 s before being removed from water vessel.

### Zeta potential measurement

The IL-modified electrocatalysts were used to prepare ink solutions for zeta potential measurements. Ink samples were prepared by adding 0.1 wt% carbon in solution using a 5:5 weight ratio of IPA and DI water. The samples were then sonicated for roughly 30 min or until a homogenous mixture was obtained. For ink dispersions containing ionomer, a constant I/C of 0.8 was maintained using Nafion Dispersion D521. Zeta potential measurements were performed using Horiba SZ-100 and a count rate of 100 kCPS to ensure the samples were sufficiently dilute for electrophoretic measurements. In order to avoid bubbles interfering with the measurements during pipetting, the ink solution into the electrode cell, the cell was tilted or tapped slowly to force the bubble out. Once the electrode was sufficiently filled with the sample, parameters including refractive index, viscosity and dielectric constant were specified within the software for the given dispersion medium.

### X-ray photoelectron spectroscopy (XPS)

XPS was performed on a Kratos AXIS Supra spectrometer with a monochromatic Al Kα source. For CO (MATHESON HG G1918775, ultra-high purity) dosing experiment, samples were moved to a chamber at the back of the Surface Science Station (SSS) inside the Kratos AXIS Supra spectrometer. The pressure of the chamber is the same as the pressure in the SSS. Before dosing the CO gas, gas lines were vacuumed to at least $6 \times 10^{-9}$ Torr. CO flowed into the chamber until the pressure reached $2 \times 10^{-5}$ Torr. When all the valves were closed (no gas exchange with outside), and the sample sited in the chamber for 10 min the chamber was vacuumed to at least $5 \times 10^{-8}$ Torr, and samples were transferred to the Specimen Analysis Chamber for measurement. Residual Gas Analyzer (MKS, e-Vision 2) was used to monitor the gas.

The powder samples were mounted on copper tape, and the MEA samples were mounted on carbon tape.

CasaXPS software was used to analyze the data. All spectra calibrated based on the sp$^2$ carbon (284 eV). 70% Gaussian/30% Lorentzian setup was used for all data but sp$^2$ carbon. An asymmetric 80% Gaussian/20% Lorentzian setup was applied for sp$^2$ carbon.

## Data availability

The data that support the findings of this study are openly available in Dryad with https://doi.org/10.7280/D1MH5Z.

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

## Acknowledgements
The authors (I.V.Z.) acknowledge funding from the National Science Foundation CAREER award 1652445. The authors acknowledge the use of facilities and instrumentation at the UC Irvine Materials Research Institute (IMRI). XPS work was performed using instrumentation funded in part by the National Science Foundation Major Research Instrumentation Program under grant no. CHE-1338173. The authors acknowledge the help of Dr. Ich Tran from IMRI during the CO dosing XPS session, and the help of Dr. Toshihiro Aoki from IMRI during the TEM session. The zeta potential measurements work was done at the HIMaC2's Analytical Laboratory, a user facility operated by the Horiba Institute for Mobility and Connectivity, University California, Irvine.

## Author contributions
A.A.: conceptualization, methodology, investigation, data curation, and writing—original draft preparation; J.L.O.: investigation; Y.H.: investigation; Y.L.: investigation; P.A.: supervision, I.V.Z.: conceptualization; funding acquisition; supervision, writing—reviewing and editing.

## Competing interests
The authors declare no competing interests.
