## [Peer review file · Nature Communications]

REVIEWER COMMENTS

Reviewer #1 (Remarks to the Author):

The manuscript “Ionic Liquids in Polymer Electrolyte Fuel Cells: New State-of-the-art in the Oxygen Reduction Reactivity and Durability” by Avid et al concerns the improvement of Pt/C fuel cell catalyst with ionic liquids (ILs). Various previous studies have demonstrated that proper IL modifications can improve ORR activity and stability, because of either higher oxygen solubility in IL or suppressed adsorption of nonreactive oxygenated species. The authors claim that these previous studies were based on RRDE techniques while a fuel cell level study is lacking.

In this study, the properties of IL are modulated by varying alkyl chain length of the cation. The physical and electrochemical properties of this series of IL-modified Pt/C catalysts were compared. The best performing IL was that achieved a tradeoff between proton conductivity and local oxygen transport resistance. It seems that the IL should have a moderate alkyl chain length. Although the authors present a thorough investigation on the structure-performance relationship, however, my overall impression is this study lacks remarkable conceptual advance in mechanism understanding.

Other major concerns are detailed as follows:

1. The distribution of IL and Nafion on the catalyst surface is dubious. The optimized IL/C ratio is 1.28, which indicates a very high usage of IL. In this case, the IL should not just fill the mesopores, but also cover the catalyst surface with a thick layer. Moreover, it is assumed that Nafion is wrapped in the outer layer of IL. However, to prepare the catalyst ink, an IL impregnated catalyst is redispersed in the Nafion solution. One can imagine that the IL would dissolve and then deposit together with Nafion after drying. The authors should provide clearer characterizations.
2. The description of the hydrophilicity of the materials is confusing. For example, on Page 6, “This indicates that ([C4dmim]+[NTf2]-) IL is the most hydrophobic IL”, “Pt/C-([C4mim]+[NTf2]-)-Nafion catalyst layer is the most hydrophobic” and “Hence, ([C4mim]+[NTf2]-) IL itself is the least hydrophobic”.
3. The degree of performance enhancement needs to be compared with the literature, at least at the RRDE level.

Minor points:

1. What does the arrow in Figure 1c, d, e point to?
2. Is there some possible IL leaching during the fuel cell operation?
3. The ink recipe may be wrong: “In order to achieve final Pt loading of 40 $\mu\text{gPt cm}^{-2}$ geo on the disk, 5 μg of the impregnated catalyst was mixed with ...”. Should the “5 μg ” read “5 mg”?
4. The full name of the abbreviation EDL (Page 10) should be provided.

Reviewer #2 (Remarks to the Author):

The manuscript presents notable research in the field of ionic liquid (IL) modified fuel cell catalyst layers. That ILs can be used to increase the activities and stability of cathodic ORR catalyst is approx. known for 10 years. But up to now transferring these lab results with catalyst immersed in aqueous phase electrolytes towards technical fuel cell membrane electrode assemblies which use a solid electrolyte (ionomer) was not possible. Especially hindering was the complex additional interplay and effect of ILs within an MEA compared to "simple" RDE studies. The presented study could overcome this limitations. First of all improved fuel cell performance also at high current densities and improved stability could be demonstrated. Furthermore, the study shows how the IL seems to interaction with mesopores, Pt, ionomer and acts as a proton transport pathway (and resistance) and as an oxygen transport resistance. Thus, viable insights to the complex interplay could be given and gives inspiring directions for future design of IL modified fuel cell catalyst layers. The manuscript is written in a clear, precise and concise manner. The experiments carried out support the statements made. It is of general interest for a broader readership and can be recommended for publication after accounting for the following points/questions:

major point: p11: "In contrast, Pt/C-([C4mim]+[NTf2]-) showed superior electrochemical properties that highlights the importance of how optimized molecular structure and induced hydrophobic nanoconfinement can alter the activity of the catalyst toward ORR"

The manuscript would benefit strongly if a more molecular based discussion of the observed differences for the three ILs would be added.

Other points:

- p2: "to 75% of Pt nanoparticles" Please clarify if w.-% is meant

- p2: "ILs within the micropores and smaller mesopores induce local nanoconfinement that can promote ORR because: i) Knudsen diffusion of oxygen taking place in the nanopores increases the frequency of oxygen molecules collision with the pore wall rather than molecule-to-molecule collision" It is unclear why more collisions with the wall despite with other oxygen molecules are advantageous.

- Figure 1 and FigureS1: What are the arrows showing? Please specify in the figure caption

- p5: "Unlike Pt/C baseline isotherm, all the IL-modified samples showed near zero nitrogen uptake at low relative pressure." At low relative pressure is an imprecise statement. Every good isotherm should show zero uptake for the lowest relative pressures. Just specify the relative pressure to be precise with the statement.

- p5 "Therefore, we can conclude that the micropores are fully impregnated with ILs and mesopores are partially filled, whereas macropores are not filled with ILs." A nice observation. It seems that the trend follows very much the trend reported for the first IL coated heterogeneous catalysts (<http://dx.doi.org/10.1002/ceat.200700050>). Can the same "simple" model be used to describe the change in pore volume like done in Fig 3 of this publication from 2007?
- p6: "Hence, ([C4mim]+[NTf2]-) IL itself is the least hydrophobic among all and has the smallest electrostatic repulsion with Nafion confirmed by zeta potential measurements" From a molecular point of view [C4mim]+[NTf2]- is more hydrophobic than [C2mim]+[NTf2]-. Thus the statement must be discussed or rephrased to point to the catalyst layer prepared with this IL and not only the IL.
- Figure 2:"by Krüss tensiometer" Krüss is a company. Probably the method and not the company should be described in the figure caption.
- Table 2: Typo "Zeta potentials zeta potential measurements"
- Table 2 seems to be a 1 to 1 repetition of Figure 2d. One should be deleted.
- p9: "Baseline Pt/C (in presence of Nafion) showed a noticeably higher proton diffusion resistance of 3.27 Ohm.cm² when compared to the catalyst layers with the mixture of Nafion and IL." A reference where to find the proton diffusion resistance should be given. Also presenting these values in a table would be good. In the actual figures the values are difficult to extract.
- p9: "MA and SA obtained from the Tafel plot in Figure 3f increased to 347 A gPt⁻¹" Here, at other positions in the text and within the tables the potential for this current needs to be given.
- p9 and tableS2: For the RDE experiments reproduction measurements with new electrodes and error bars need to be given for a high impact publication. Especially when discussing an ECSA change of 5.5 %.
- p10: "displays the optimal polarization curves obtained after two and three voltage recovery cycles" Please add the information of two/three voltage recovery cycles to the figure caption for all Figures where the procedure was applied.
- p11:"MEA level measurements revealed much higher ECSA losses compared to RDE set-up; as the modified catalysts, Pt/C-([C2mim]+[NTf2]-), Pt/C-([C4mim]+[NTf2]-) , Pt/C-([C4dmim]+[NTf2]-) lost 26.8%, 31.1%, 48.3% of the ECSA, respectively. This could be related to the molecular structure of the ILs and the interaction of the cationic alkyl chain with Pt terrace site. In correlation with previously reported observations 26, increase in the number of alkyl chains, as well as their length can increase the interaction with Pt sites" Missing in the discussion is, that through the catalyst layer formed also the accessibility Protons towards Pt can be changed through the IL, which would also to another Proton stripping result.

Reviewer #3 (Remarks to the Author):

The paper titled 'Ionic Liquids in Polymer Electrolyte Fuel Cells: New State-of-the-art in the

Oxygen Reduction Reactivity and Durability' the authors study in length the positive effects of using ionic liquids (IL) as an electrolyte promoting reactant diffusion to the buried catalyst nanoparticles and in general activity of ORR. In my opinion the paper touches on a very important research in alternative ways to fully utilize low loading PGM catalyst coatings and therefore to make the hydrogen economy a reality. The topic of using IL to fill the normally unutilized support pores in the catalyst layer is relatively new and any information regarding the fundamental processes leading to better utilization is welcomed for further development. The paper is clearly written and the choice of analytical techniques, data analysis and interpretation is sound. Therefore I think that given the novelty and importance of the findings presented in this paper, and the topic in general, the publication of this work in Nature Communication is fully justified. I recommend publishing this paper after considering a few minor comments.

Pg2_line48-49 : Authors claim that the carbon support dictates the stability and performance of the catalyst. While certainly the support plays a role in the stability and performance, it does not have a major influence. There is vast literature about the stability and performance drivers of the catalyst layer.

Pg5_line134-136: The sentence starting "Difference was seen..." is hard to understand. Do authors mean that for both mesopores and micropores there is the difference compared to the baseline? Or is there also a difference between different ILs?

Pg9-10_line210-220: This is interesting. Presumably the IL protects the active sites from oxidizing, improving the activity of the catalyst. But this should be true for all studied IL as they all show hydrophobicity. So why only the C4min show better performance to the Pt baseline? Perhaps, as mentioned later in the paper, this could be due to the IL interaction with the surface? Some ILs also poison the active sites?

Pg10_line219-220: improved compared to what.. Baseline?

Pg11_line256: The authors could mention here the IL/C ratio for the samples used for in-situ characterization.

Pg13_line314-316: Do not see 100%RH in the plot S7. Can the authors add this for the comparison?

REVIEWER COMMENTS

Reviewer #1:

The manuscript “Ionic Liquids in Polymer Electrolyte Fuel Cells: New State-of-the-art in the Oxygen Reduction Reactivity and Durability” by Avid et al concerns the improvement of Pt/C fuel cell catalyst with ionic liquids (ILs). Various previous studies have demonstrated that proper IL modifications can improve ORR activity and stability, because of either higher oxygen solubility in IL or suppressed adsorption of nonreactive oxygenated species. The authors claim that these previous studies were based on RRDE techniques while a fuel cell level study is lacking.

In this study, the properties of IL are modulated by varying alkyl chain length of the cation. The physical and electrochemical properties of this series of IL-modified Pt/C catalysts were compared. The best performing IL was that achieved a tradeoff between proton conductivity and local oxygen transport resistance. It seems that the IL should have a moderate alkyl chain length.

1- Although the authors present a thorough investigation on the structure-performance relationship, however, my overall impression is this study lacks remarkable conceptual advance in mechanism understanding.

Response: We highly appreciate the reviewer’s insightful comments on this manuscript. As the reviewer suggested that more profound discussion on the mechanisms is needed, we added the following statements to the introduction and section 2.2. ILs’ proton activity at Pt|IL interface and its effect on ORR is discussed in more details from a molecular structure point of view. We also added a discussion to the RDE section of the manuscript on how molecular structure of ILs influence their electrochemical properties. In addition, to elaborate on IL roles in improving the activity under in-situ experiment, we included the CO stripping data to the supplementary information (Figure S9) and discussed how ligand effect can influence the Pt surface properties.

Changes to the manuscript:

In introduction section:

“Alkyl imidazolium bis- (trifluoromethanesulfonyl)imide ILs have met the criteria of suitable modifier for fuel cell catalyst design due to their high ionic conductivity and oxygen solubility, superior thermal and electrochemical stability under fuel cell tested temperature and voltage, low melting point, viscosity, and vapor pressure. They are also non-flammable and have high ΔpK_a value which is shown to be sufficient for satisfactory proton transfer^{24,31,32,33}. It has been shown that the pK_a of the cations in the IL dictates the local proton activity near Pt active sites. Technically, by the formation of hydrogen-bond network at the Pt|IL interface, the cation of IL would function as a proton donor facilitating the kinetics of the reaction³⁴.”

In Results and discussion section 2.2: The structure-property relationship of these three ILs originates from both their interactions with Pt surface and induced hydrophobic microenvironment²⁷. Elongation of alkyl chain can result in stronger interactions with Pt terrace sites. As shown in CV voltammograms and Table S3, by elongation of side chain from C₂ in Pt/C-([C₂mim]⁺[NTf₂]⁻) to C₄ in Pt/C-([C₄mim]⁺[NTf₂]⁻), the ECSA decreased indicating that butyl chains have blocked some Pt active sites. However, side chain interactions with those active sites make them less prone to formation of oxygenated species²⁷. Pt/C-([C₄dmim]⁺[NTf₂]⁻) has an additional interactive methyl group and showed super hydrophobic properties compared to the other two ILs; therefore, it provides the most dominant suppression of Pt-oxide formation. [C₄dmim]⁺[NTf₂]⁻ is the most effective IL in protecting Pt active sites from being oxidized by repelling water molecules and [C₂mim]⁺[NTf₂]⁻ has the least ECSA drop and passivation of Pt active sites. However, from polarization curves displayed in **Error! Reference source not found.b** it can be noted that only Pt/C-([C₄mim]⁺[NTf₂]⁻) catalyst film demonstrated a potential improvement in ORR kinetics as the half-wave potential (E_{1/2}) is slightly shifted towards higher potential compared to the baseline.”

“Furthermore, it has been shown that ORR activity enhancement using ILs is also attributed to the interfacial hydrogen-bonding between IL’s cation and adsorbed intermediate oxygenated species of ORR (O_{ad} and OH_{ad}). In this study, N-H⁺ bonds of the imidazolium cation form hydrogen-bonding with OH_{ad}; therefore, it enables faster proton tunneling from IL to Pt active sites through intermediate species (N-H⁺ + OH_{ad} + e⁻ → H₂O + N). Hydrogen-bonding structure can regulate the exchange current density of ORR and reaction rate constant based on the vibrational features of the intermediates³⁴.”

In Results and discussion section 2.3:in explanation of imidazolium groups interaction with Pt:

“The oxidative peak observed during IL-induced displacement of CO on Pt is attributed to the imidazolium groups adsorbing on surface. Sabota et al.³⁹ previously have shown that imidazolium cations have electronic interactions with Pt through π-system of the imidazolium and the formation of carbene species at the surface. This observation can be further confirmed with our CO stripping data shown in Figure S9. The negative shift of CO stripping peak to lower potentials indicates a weaker Pt-CO_{ads} bonding in presence of ILs due to the ligand-like interactions. Therefore, imidazolium-derived ILs significantly modify Pt d-band centers via ligand effects. During the co-adsorption of IL with CO, IL molecules preferentially interact with Pt(111) terrace sites, while CO selectively adsorbs on the edges. It can be seen in Figure S9 that ([C₄mim]⁺[NTf₂]⁻) shows the smallest left shift compared to the other two ILs. One can comprehend that ([C₄mim]⁺[NTf₂]⁻) can induce the strongest bonding between Pt and reactant molecules (O₂ and CO) compared to the other two IL⁹; while it effectively prevent SO₃⁻ groups adsorption onto the surface of Pt, adding to its catalytic activity.”

Other major concerns are detailed as follows:

2- The distribution of IL and Nafion on the catalyst surface is dubious. The optimized IL/C ratio is 1.28, which indicates a very high usage of IL. In this case, the IL should not just fill the mesopores, but also cover the catalyst surface with a thick layer. Moreover, it is assumed that Nafion is wrapped in the outer layer of IL. However, to prepare the catalyst ink, an IL impregnated catalyst is redispersed in the Nafion solution. One can imagine that the IL would dissolve and then deposit together with Nafion after drying. The authors should provide clearer characterizations.

Response: We thank the reviewer for raising these interesting points. For IL loading determination, we referred to the paper by Huang et al. (Huang, K., Song, T., Morales-Collazo, O., Jia, H. & Brennecke, J. F. Enhancing Pt/C Catalysts for the Oxygen Reduction Reaction with Protic Ionic Liquids: The Effect of Anion Structure. *J. Electrochem. Soc.* **164**, F1448–F1459 (2017).) They used TEC10E50E, 50 wt.% Pt supported on high surface area carbon black with specific surface area $800 \text{ m}^2 \text{ g}^{-1}$. The catalyst we used was 40wt.% Pt supported on high surface area Ketjenblack EC-300J, with approximately the same specific area. Huang et. al showed in their study that the optimized ratio of IL to carbon is 2.56 in RDE set-up; therefore, we relied on this ratio and performed RDE and MEA electrochemical characterizations. Based on our MEA results, we observed that the optimum amount of IL/C should be 1.28. (please see section 2.2., RDE prescreening and IL/C ratio optimization in MEA).

In Huang et. al study, the morphology of IL on Pt/C was shown using TEM. They claimed that for IL/C ratio of 2.56, a thin layer of IL has covered Pt/C surface. On the other hand, in our BET study we used a carbon-supported catalyst with similar carbon specific surface area but with IL/C ratio of 1.28. Since the amount of IL has been cut into half, we believe the pore filling degree shown by the BET is reliable. Although we detected some lumps of IL on the surface through TEM images, our BET results confirm the successful impregnation of IL in micropores and mesopores.

Regarding reviewer's comment about IL redispersion in the ink solution, we performed an experiment as follows:

First of all, we prepared impregnated Pt/C powder with all three ILs. Then, we weighed 100 mg of each powder and mixed it with 2 ml of IPA and DI water and ultrasonicated it for 30 mins. 100 mg of IL-impregnated powder includes about 43.75 mg of IL considering the IL/C ratio of 1.28. In the next step, we centrifuged the suspensions at 30,000 rpm for 15 mins until the powder separated completely from the solvents, as can be seen below. We removed the solvent and let the powders dry overnight. Then we weighed all the powders after the procedure. The weight difference between the original IL-impregnated powders and the ones after the procedure can be attributed to the ILs that dissolved in the solution, as they are soluble in the mixture of IPA and water. Weight difference for Pt/C-([C₂mim]⁺[NTf₂]⁻), Pt/C-([C₄mim]⁺[NTf₂]⁻), and Pt/C-([C₄dmim]⁺[NTf₂]⁻), were 18.1, 22.1, and 24.8 mg, respectively. We also performed this procedure on a control sample in which we saw about 10.6 mg weight loss. This loss is due to the potential error in the experiment as we remove the separated solution from the precipitated powder. We can say that by this procedure Pt/C-([C₂mim]⁺[NTf₂]⁻), Pt/C-([C₄mim]⁺[NTf₂]⁻), and Pt/C-

$[\text{C}_4\text{dmim}]^+[\text{NTf}_2]^-$) lost about 17, 26, and 32 wt.% of the IL (they dissolved in the IPA and water mixture and got separated from the powder). This means that a part of the ILs in the impregnated powders redispersed in the solution and redeposited together with Nafion. We believe, IL molecules on catalyst surface, in the macropores and bigger mesopores would dissolve faster compared to the ones in the smaller mesopores and micropores. This observation is in great agreement with the IL molecules chemical structures and BET results. As can be seen, the ILs dissolution in IPA and water increases with elongation in alkyl chain length. We mentioned in the manuscript that “ $[\text{C}_4\text{dmim}]^+[\text{NTf}_2]^-$) has more complex structure as it has both butyl and methyl cationic chains, higher free volume, and less packing of the molecules in the pores as a result.” Therefore, the more complex the structure of IL is, the less it will fill the pores and easily get redispersed in the solution.

Based on these observations, we confirm that reviewer’s point is correct and during ink preparation for MEA fabrication, IL molecules redisperse in the solution and redeposit with Nafion after drying.

Changes to the manuscript:

We modified the manuscript to indicate this observation in MEA optimization section (section 2.3) as follows:

“In order to prepare the ink for MEA fabrication, impregnated powders were redispersed in the Nafion solution in water and IPA. The ink suspension was then coated, dried, and hot-pressed on Nafion membrane for MEA manufacture as described in Materials and Methods section 3.4. To investigate the ILs evolution during MEA preparation, we performed a complementary experiment. Details of this experiment and observations are summarized in the Supporting Information. We believe that depending on the IL chemical structure and pore filling degree, a part of IL that is impregnated in the catalyst powder would redissolve in Nafion solution as IL is soluble in the mixture of water and IPA. Then, it deposits along with Nafion during coating and drying.

One can assume that IL molecules on the surface, in the macropores and bigger mesopores would dissolve faster in the Nafion solution compared to the ones packed in the micropores and smaller mesopores and deposit together with Nafion in decal.”

In addition, we explained the optimum amount of IL in the pores would be $IL/C=1.28$, based on this discussion at the end of section 2.3.

“The reason we believe 1.28 is the optimum amount for MEA fabrication is that based on our study summarized in the supporting information, less than 30% of the IL is covering the surface or is in the bigger pores. Therefore, about 70% of the IL is deposited in the micro and smaller mesopores and would not dissolve as fast the ones on the surface and would stay in the smaller pores facilitating proton transfer. On the other hand, at high IL loadings ($IL/C=2.56$), there is a layer of IL on the surface that would redeposit with Nafion in the catalyst layer causing high ionic resistance.”

Experiment procedure described above is also added to the “evolution of Pt/C-IL interface during MEA fabrication procedure” section of the supplementary information.

3- The description of the hydrophilicity of the materials is confusing. For example, on Page 6, “This indicates that ([C4dmim]⁺[NTf₂]⁻) IL is the most hydrophobic IL”, “Pt/C-([C4mim]⁺[NTf₂]⁻)-Nafion catalyst layer is the most hydrophobic” and “Hence, ([C4mim]⁺[NTf₂]⁻) IL itself is the least hydrophobic”.

Response: We tried our best to carefully rewrite and reorganize this section to improve the logic flow as follows:

Change to the manuscript:

“By comparing the amount of water absorbed by the samples during the first 900s while they are immersed into the water, it can be concluded that baseline Pt/C coated catalyst layer shows the least amount of water uptake of about 18 mg compared to all IL-containing samples. On the other hand, all IL-containing catalyst layers show higher amount of water adsorption indicating that all Pt/C-IL-Nafion films are more hydrophilic compared to baseline. For instance, Pt/C-([C4dmim]⁺[NTf₂]⁻)-Nafion catalyst layer has the highest water uptake amount of all (57 mg); hence, it is the most hydrophilic catalyst layer. Here we can illustrate with a schematic the potential intramolecular interactions between the IL molecules and Nafion domains (Figure S2). IL molecules are prone to induce a hydrophobic microenvironment that repels the SO₃⁻ groups of Nafion to the outer surface resulting in more water absorption to the catalyst layer. Therefore, one can assume that the more hydrophobic the IL molecules are, the more hydrophilic the catalyst layer along with the Nafion would be. External contact angles shown in Table 1 were calculated using Wilhelmy equation and sudden increase in the water mass gain after each sample was removed

from the water (Δmg). The details of experiment can be found in the Materials and Methods section of this paper. Pt/C-([C₄dmim]⁺[NTf₂]⁻)-Nafion film exhibited the smallest external contact angle of 137° and the highest amount of water uptake of 57 mg among all IL-containing samples. This indicates that ([C₄dmim]⁺[NTf₂]⁻) IL molecule itself is the most hydrophobic IL of all three and induced the strongest electrostatic repulsion on SO₃⁻ groups of Nafion to reorient to the outer surface. These results are in agreement with zeta potential measurements in which hydrophobic interactions between the IL molecules and Nafion's PTFE backbone possibly dictate the conformational structure of the Nafion polymer chains. A stronger electrostatic repulsion between the molecules of ILs and Nafion's PTFE backbone results in a higher local charge density at the interface, and larger zeta potential ("Figure 3c and 2d). From tensiometry results, we observed that Pt/C-([C₄mim]⁺[NTf₂]⁻)-Nafion catalyst layer has the largest external contact angle of 148° and the least water uptake of about 33 mg; therefore, it is the most hydrophobic catalyst layer compared to the other two IL-containing catalyst layers. Hence, ([C₄mim]⁺[NTf₂]⁻) IL molecule itself induces the least hydrophobic microenvironment compared to the other two ILs and has the smallest electrostatic repulsion with Nafion chains confirmed by zeta potential measurements. "

4- The degree of performance enhancement needs to be compared with the literature, at least at the RRDE level.

Response: In our previously published review paper Avid, A. & Zenyuk, I. V. "Confinement effects for nano-electrocatalysts for oxygen reduction reaction". *Curr. Opin. Electrochem.* 25, 100634 (2021).), we have summarized the ORR performance enhancement with small molecule modification in RDE setup.

Change to manuscript:

We added following statements to the manuscript as a comparison to the literature:

"There have been extensive investigations in the literature to show ORR improvement in IL-modified Pt/C catalysts using RDE²⁴. Huang et al.²⁹ indicated MA enhancement factor of about 1.2 for both [MTBD][beti] and [MTBD][C₄F₉SO₃]-modified high surface area Pt/C. These samples improved SA by a factor of 1.4 and 1.7. The degree of performance enhancement is in a great agreement with our observations in which the MA and SA enhancement factors for high surface area Pt/C-([C₄mim]⁺[NTf₂]⁻) were 1.2 and 1.7, respectively. Zhang et al.²⁷ explored the ORR enhancement on low surface area Pt/C using imidazolium based ILs and obtained MA enhancement factor of about 3 for their best performing IL that has similar chemical structure to ([C₄mim]⁺[NTf₂]⁻) and has the same cationic alkyl chain length. Yan et. al.³⁰ also reported up to three-fold enhancement in MA and SA of Pt/C catalyst by replacing Nafion with a poly(ionic liquid) in which the IL segment acts as a proton conductor."

Minor points:

5- What does the arrow in Figure 1c, d, e point to?

Response: Arrows show the lumps of ionic liquids.

Change to the manuscript: We added a description to Figure 1 and Figure S1 captions.

Figure 1 caption:

“Figure 1. a) schematic of HSA Pt/C catalysts with micropores filled with either water or imidazolium-derived ILs, TEM images of b) pristine 40 wt.% HSA Ketjenblack Pt/C obtained from Fuel Cell Store, c) Pt/C-([C₂mim]⁺[NTf₂]⁻), d) Pt/C-([C₄mim]⁺[NTf₂]⁻), e) Pt/C-([C₄dmim]⁺[NTf₂]⁻). Arrows show the ionic liquids lumps. “

Figure S1 caption:

“Figure S1. TEM images of a, b, c) Baseline (pristine) 40 wt. % Pt/C obtained from Fuel Cell Store, IL-modified Pt/C catalysts with d, e, f) Pt/C-([C₂mim]⁺[NTf₂]⁻), g, h, i) Pt/C-([C₄mim]⁺[NTf₂]⁻), and j, k, l) Pt/C-([C₄dmim]⁺[NTf₂]⁻). Arrows show the ionic liquids lumps. “

6- Is there some possible IL leaching during the fuel cell operation?

Response: Yes, we think IL molecules will leach out of the catalyst layer during potential cycling, specifically the ones that are in the larger mesopores or on the surface. In addition, during voltage recovery cycles in which the cell was kept at low potential (0.2 V) and high RH (150%), water can wash out IL from the catalyst layer. This is the reason that MEA prepared with higher loading of IL (IL/C ratio of 2.56) needed 8 voltage recovery cycles, while MEA with half of the loading needed only 3 voltage recovery cycles to achieve the optimum performance.

Changes to the manuscript: None.

7- The ink recipe may be wrong: “In order to achieve final Pt loading of 40 μgPt cm⁻²geo on the disk, 5 μg of the impregnated catalyst was mixed with ...”. Should the “5 μg” read “5 mg”?

Response: We thank the reviewer for catching this point. We corrected the recipe.

Change to the manuscript:

Section 3.3.1:

“In order to achieve final Pt loading of 40 μgPt cm⁻²geo on the disk, 5 mg of the impregnated catalyst was mixed with 996.5 μl of MilliQ H₂O:IPA (24 wt.% IPA), and 8.12 μl of a 5 wt.% Nafion solution.”

8- The full name of the abbreviation EDL (Page 10) should be provided.

Response: Since the double layer capacitance was first mentioned in the introduction, we added the abbreviation to introduction instead.

Change to the manuscript:

Introduction:

“ILs within the micropores and smaller mesopores induce local nanoconfinement that can promote ORR because: *i*) Knudsen diffusion of oxygen taking place in the nanopores increases the frequency of oxygen molecules collision with the catalyst surface sites rather than molecule-to-molecule collision, *ii*) electric double layers (EDLs) are thin for ILs|Pt interface compared to water|Pt interface²⁴.”

Reviewer #2 (Remarks to the Author):

The manuscript presents notable research in the field of ionic liquid (IL) modified fuel cell catalyst layers. That ILs can be used to increase the activities and stability of cathodic ORR catalyst is approx. known for 10 years. But up to now transferring these lab results with catalyst immersed in aqueous phase electrolytes towards technical fuel cell membrane electrode assemblies which use a solid electrolyte (ionomer) was not possible. Especially hindering was the complex additional interplay and effect of ILs within an MEA compared to "simple" RDE studies. The presented study could overcome this limitations. First of all improved fuel cell performance also at high current densities and improved stability could be demonstrated. Furthermore, the study shows how the IL seems to interaction with mesopores, Pt, ionomer and acts as a proton transport pathway (and resistance) and as an oxygen transport resistance. Thus, viable insights to the complex interplay could be given and gives inspiring directions for future design of IL modified fuel cell catalyst layers. The manuscript is written in a clear, precise and concise manner. The experiments carried out support the statements made. It is of general interest for a broader readership and can be recommended for publication after accounting for the following points/questions:

9- major point: p11: "In contrast, Pt/C-([C4mim]+[NTf2]-) showed superior electrochemical properties that highlights the importance of how optimized molecular structure and induced hydrophobic nanoconfinement can alter the activity of the catalyst toward ORR" The manuscript would benefit strongly if a more molecular based discussion of the observed differences for the three ILs would be added.

Response:

We highly appreciate the reviewer for their helpful comments on our manuscript. We have thoroughly discussed all three ILs interactions with Nafion chains from molecular point of view in section 2.1 using zeta potential and water uptake measurements. Repulsive forces between IL

molecules and Nafion chains dictate the hydrophobicity degree of the CL. We added a discussion to the RDE section of the manuscript on how molecular structure of ILs influence their electrochemical properties. In addition, to elaborate on IL roles in improving the activity under in-situ experiment, we included the CO stripping data to the supplementary information (Figure S9) and discussed how ligand effect can influence the Pt surface properties.

Changes to the manuscript:

Results and discussion section 2.2:

“The structure-property relationship of these three ILs originates from both their interactions with Pt surface and induced hydrophobic microenvironment²⁷. Elongation of alkyl chain can result in stronger interactions with Pt terrace sites. As shown in CV voltammograms and Table S3, by elongation of side chain from C₂ in Pt/C-([C₂mim]⁺[NTf₂]⁻) to C₄ in Pt/C-([C₄mim]⁺[NTf₂]⁻), the ECSA decreased indicating that butyl chains have blocked some Pt active sites. However, side chain interactions with those active sites make them less prone to formation of oxygenated species²⁷. Pt/C-([C₄dmim]⁺[NTf₂]⁻) has an additional interactive methyl group and showed super hydrophobic properties compared to the other two ILs; therefore, it provides the most dominant suppression of Pt-oxide formation. [C₄dmim]⁺[NTf₂]⁻ is the most effective IL in protecting Pt active sites from being oxidized by repelling water molecules and [C₂mim]⁺[NTf₂]⁻ has the least ECSA drop and passivation of Pt active sites. However, from polarization curves displayed in **Error! Reference source not found.** it can be noted that only Pt/C-([C₄mim]⁺[NTf₂]⁻) catalyst film demonstrated a potential improvement in ORR kinetics as the half-wave potential (E_{1/2}) is slightly shifted towards higher potential compared to the baseline.”

Results and discussion section 2.3: in explanation of imidazolium groups interaction with Pt:

“The oxidative peak observed during IL-induced displacement of CO on Pt is attributed to the imidazolium groups adsorbing on surface. Sabota et al.³⁹ previously have shown that imidazolium cations have electronic interactions with Pt through π -system of the imidazolium and the formation of carbene species at the surface. This observation can be further confirmed with our CO stripping data shown in Figure S9. The negative shift of CO stripping peak to lower potentials indicates a weaker Pt-CO_{ads} bonding in presence of ILs due to the ligand-like interactions. Therefore, imidazolium-derived ILs significantly modify Pt d-band centers via ligand effects. During the CO displacement in presence of ILs, IL molecules preferentially interact with Pt(111) terrace sites, while CO selectively adsorbs on the edges. It can be seen in Figure S9 that ([C₄mim]⁺[NTf₂]⁻) shows the smallest left shift compared to the other two ILs. One can comprehend that ([C₄mim]⁺[NTf₂]⁻) can induce the strongest bonding between Pt and reactant molecules (O₂ and CO) compared to the other two IL⁹; while it effectively prevent SO₃⁻ groups adsorption onto the surface of Pt, adding to its catalytic activity.”

Other points:

10- p2: "to 75% of Pt nanoparticles" Please clarify if w.-% is meant.

Response: This is a great point. High surface area carbon support can host up to 75% of the Pt surface area within the wide mesopores.

Change to the manuscript: We rephrased this sentence in introduction to clarify this description.

“High surface area carbon support is considered to be a great candidate for fuel cell application owing to its high internal mesoporosity. It can host up to 75% of Pt active surface area within the wide mesopores with size between 5.5 and 14 nm.”

11- p2: "ILs within the micropores and smaller mesopores induce local nanoconfinement that can promote ORR because: i) Knudsen diffusion of oxygen taking place in the nanopores increases the frequency of oxygen molecules collision with the pore wall rather than molecule-to-molecule collision" It is unclear why more collisions with the wall despite with other oxygen molecules are advantageous.

Response: Based on the molecule/surface interaction mechanisms that take place in the Knudsen regime, the average time interval between oxygen molecules collision with the catalyst surface site determines the ORR rate. As the catalysts are located within the mesopores in high surface area carbon supports, in the Knudsen regime, the increased frequency of molecular collision with the pore wall (on which the catalyst nanoparticles are deposited) will promote the ORR.

Change to the manuscript:

In the corresponding sentence, we changed “pore wall” to “catalyst surface site” for better clarification.

“ILs within the micropores and smaller mesopores induce local nanoconfinement that can promote ORR because: i) Knudsen diffusion of oxygen taking place in the nanopores increases the frequency of oxygen molecules collision with the catalyst surface sites rather than molecule-to-molecule collision.”

12- Figure 1 and FigureS1: What are the arrows showing? Please specify in the figure caption

Response: Arrows show the lumps of ionic liquids.

Change to the manuscript: *We added a description to Figure 1 and Figure S1 captions.*

Figure 1 caption:

“Figure 2. a) schematic of HSA Pt/C catalysts with micropores filled with either water or imidazolium-derived ILs, TEM images of b) pristine 40 wt.% HSA Ketjenblack Pt/C obtained

from Fuel Cell Store, c) Pt/C-([C₂mim]⁺[NTf₂]⁻), d) Pt/C-([C₄mim]⁺[NTf₂]⁻), e) Pt/C-([C₄dmim]⁺[NTf₂]⁻). Arrows show the ionic liquids lumps. “

Figure S1 caption:

“Figure S1. TEM images of a, b, c) Baseline (pristine) 40 wt. % Pt/C obtained from Fuel Cell Store, IL-modified Pt/C catalysts with d, e, f) Pt/C-([C₂mim]⁺[NTf₂]⁻), g, h, i) Pt/C-([C₄mim]⁺[NTf₂]⁻), and j, k, l) Pt/C-([C₄dmim]⁺[NTf₂]⁻). Arrows show the ionic liquids lumps. “

13- p5: "Unlike Pt/C baseline isotherm, all the IL-modified samples showed near zero nitrogen uptake at low relative pressure." At low relative pressure is an imprecise statement. Every good isotherm should show zero uptake for the lowest relative pressures. Just specify the relative pressure to be precise with the statement.

Change to the manuscript:

“Unlike Pt/C baseline isotherm with a significant nitrogen uptake of $\sim 100 \text{ cm}^3 \text{ g}^{-1}$, all three IL-modified samples showed near zero nitrogen uptake at relative pressure below 0.01 ($p/p_0 < 0.01$). However, at high relative pressure ($p/p_0 > 0.9$), their sorption isotherm curves showed similar shape to that of the pristine Pt/C sample as indicated in Figure 2a.”

14- "p5 "Therefore, we can conclude that the micropores are fully impregnated with ILs and mesopores are partially filled, whereas macropores are not filled with ILs." A nice observation. It seems that the trend follows very much the trend reported for the first IL coated heterogeneous catalysts (<http://dx.doi.org/10.1002/ceat.200700050>). Can the same "simple" model be used to describe the change in pore volume like done in Fig 3 of this publication from 2007?

Response:

We thank the reviewer for raising this interesting point. However, this model was only good for predicting the pore filling for large pores and at low IL loading. From figure 3 in the mentioned paper, we can see that the model went off trends for the N₂ physisorption points even at low IL loading and went significantly off trend through the whole range at high IL loading. The author also mentioned the limitation of this mode for mesopores between 2nm and 50nm, which is the focusing range of our paper. But instead, we further decipher the local IL filling and distribution in mesopore based on our N₂ physisorption model. We have updated the pore structure discussion and given a more specific description on the occupation of micro-, meso- and macro-pores by ionic liquids.

Change to the manuscript:

“Figure 3a demonstrates the nitrogen physisorption isotherms for baseline Pt/C and all IL-modified samples with IL/C of 1.28, at 77 K. Unlike Pt/C baseline isotherm with a significant nitrogen uptake of $\sim 100 \text{ cm}^3 \text{ g}^{-1}$, all three IL-modified samples showed near zero nitrogen uptake at relative

pressure below 0.01 ($p/p_0 < 0.01$). However, at high relative pressure ($p/p_0 > 0.9$), their sorption isotherm curves showed similar shape to that of the pristine Pt/C sample as indicated in Figure 2a. As illustrated by the log-scale isotherm in Figure 2a, the nitrogen uptake starting points for IL-modified samples were around 0.01 P/P_0 , indicating a complete filling or blocking of sub-micropores below 1 nm. Pore size distribution as a function of pore diameter is shown in Figure 3b in order to assess the filling degree and IL coverage of the IL-containing Pt/C powders. For pores with diameter above 50 nm, almost no difference was observed between the pore volume of the baseline and three IL-modified samples indicating that the macropores of the modified powders are not filled with ILs. For pores between 2 nm to 50 nm, the pore volume decreased at least 50% showing that all IL molecules have partially filled mesopores. On the other hand, over 90% drop in the pore volume is observed for micropores between 1 nm and 2 nm indicating that the micropores are fully impregnated with ILs. This is in agreement with the significantly reduced nitrogen uptake below 0.2 P/P_0 indicated in Figure 2a. For the pore size between 2 nm and 20 nm, Pt/C-([C₄mim]⁺[NTf₂]⁻) samples showed non-negligible difference as compared to the other two IL-modified samples (Figure 2b), where its pore volume was still well below the Pt/C baseline but higher than that of its Pt/C-([C₂mim]⁺[NTf₂]⁻) and -([C₄mim]⁺[NTf₂]⁻) counterparts. This is partially due to the more complex structure of the [C₄mim]⁺, as it has both butyl and methyl cationic chains, higher free volume, and less packing of the molecules in the pores as a result. For interface modification in this work, the small mesopores are of more relevance as Pt most likely cannot be deposited into micropores and the ionomer cannot access small mesopores below <20 nm. From BET observations and TEM images in Figure 1, one can conclude that most of the Pt nanoparticles were well covered by the ILs.

15- p6: "Hence, ([C₄mim]⁺[NTf₂]⁻) IL itself is the least hydrophobic among all and has the smallest electrostatic repulsion with Nafion confirmed by zeta potential measurements" From a molecular point of view [C₄mim]⁺[NTf₂]⁻ is more hydrophobic than [C₂mim]⁺[NTf₂]⁻. Thus the statement must be discussed or rephrased to point to the catalyst layer prepared with this IL and not only the IL.

Change to the manuscript:

"From tensiometry results, we observed that Pt/C-([C₄mim]⁺[NTf₂]⁻)-Nafion catalyst layer has the largest external contact angle of 148° and the least water uptake of about 33 mg; therefore, it is the most hydrophobic catalyst layer compared to the other two IL-containing catalyst layers. Hence, ([C₄mim]⁺[NTf₂]⁻) IL molecule itself induces the least hydrophobic microenvironment compared to the other two ILs and has the smallest electrostatic repulsion with Nafion chains confirmed by zeta potential measurements."

16- Figure 2: "by Krüss tensiometer" Krüss is a company. Probably the method and not the company should be described in the figure caption.

Response: We updated the caption by mentioning “capillary penetration method” instead of company name.

Change to the manuscript:

“Figure 3. a,b) Nitrogen physisorption at 77 K, c) mass evolution of catalyst layers containing various ILs with IL/C ratio of 1.28 measured using capillary penetration method. d) zeta potential measurements for pristine and IL-modified Pt/C with and without Nafion.”

17- Table 2: Typo "Zeta potentials zeta potential measurements" –

Response: Corrected.

18- Table 2 seems to be a 1 to 1 repetition of Figure 2d. One should be deleted.

Change to the manuscript: *This table is moved to the supplementary information as Table S1.*

19- p9: "Baseline Pt/C (in presence of Nafion) showed a noticeably higher proton diffusion resistance of 3.27 Ohm.cm² when compared to the catalyst layers with the mixture of Nafion and IL." A reference where to find the proton diffusion resistance should be given. Also presenting these values in a table would be good. In the actual figures the values are difficult to extract.

Response: The values for R_{H^+} are summarized in Table S3.

Change to the manuscript: *We added this sentence to section 2.2 referring to the supplementary Information.*

“The values for R_{H^+} can be found in Table S3.”

20- p9: "MA and SA obtained from the Tafel plot in Figure 3f increased to 347 A gPt⁻¹" Here, at other positions in the text and within the tables the potential for this current needs to be given.

Response: All mass and specific activities were measured at 0.9 V.

Change to the manuscript: *We have added this information to the text and all the tables (Table 2, Table S3 and S4) in the manuscript and supplementary information.*

Section 2.2:

“All mass and specific activities were measured at 0.9 V.

Although the SA at 0.9 V increased compared to the baseline, the MA was only 110 A gPt⁻¹. By cutting the amount of IL into half, IL/C ratio of 1.28, MEA showed enhanced performance across all current densities, with optimum peak power density of 0.909 W cm⁻². MA and SA obtained

from the Tafel plot at 0.9 V in Error! Reference source not found.f increased to 347 A g_{Pt}⁻¹ and 697 μA cm²_{Pt}, respectively. The amount of ([C₄mim]⁺[NTf₂]⁻) was then reduced to half, causing the MA and SA at 0.9 V drop to 173 A g_{Pt}⁻¹ and 509 μA cm²_{Pt}.”

Section 2.3:

“However, in this case, the notable loss in the active area is responsible for the significant drop in the catalytic activity as the MA obtained at 0.9 V decreased to 121 A g_{Pt}⁻¹ (Table 2 and “Figure 5a).

As illustrated in “Figure 5a and b and Table 2, Pt/C-([C₄mim]⁺[NTf₂]⁻) showed enhanced mass activity and specific activity at 0.9 V of approx. 350 A g_{Pt}⁻¹ and 700 (μA cm²_{Pt}) which is 20% and 75% higher than that of the baseline.”

21- p9 and tableS2: For the RDE experiments reproduction measurements with new electrodes and error bars need to be given for a high impact publication. Especially when discussing an ECSA change of 5.5 %.

Change to the manuscript: the experiment reproduction data with new electrodes is shown in Table S2 for each IL-containing Pt/C.

22- p10: "displays the optimal polarization curves obtained after two and three voltage recovery cycles" Please add the information of two/three voltage recovery cycles to the figure caption for all Figures where the procedure was applied.

Response: The number of voltage recovery cycles for all fuel cell tested MEAs was added to Table 2, Figure 4,5, and 6.

Change to the manuscript:

“Table 2. Electrochemical properties of baseline Pt/C MEA obtained after two voltage recovery cycles and Pt/C-IL MEAs obtained after three voltage recovery cycles in fuel cell set-up.”

“Figure 4. Electrochemical characterization of IL-containing MEAs obtained after three voltage recovery cycles vs. baseline Pt/C after two voltage recovery cycles evaluated at 80°C and 100% RH: a) H₂/air polarization curves, b) power density at 150 kPa_{abs} total pressure, c) cyclic voltammograms recorded in H₂/N₂ and 100 kPa_{abs} total pressure, d) H₂/O₂ Tafel plots at 150 kPa_{abs} total pressure, e) CO-displacement at 0.35 V at 100 kPa_{abs} and 40°C, f) Correlation between local oxygen mass transport resistance and proton conductivity in 5 cm² differential cells, cathode: HSA Pt/C or HSA Pt/C-IL, anode: LSA Pt/C. “

“Figure 5. Electrochemical characterization of IL-containing MEAs obtained after three voltage recovery cycles vs. baseline Pt/C obtained after two voltage recovery cycles evaluated at 80°C:

a) mass and b) specific activities at 0.9 V, 150 kPa_{abs} and 100% RH, and c) local oxygen transport resistance measured at 75% RH, and d) ionic conductivity at 100% RH. “

“Figure 6. a) comparison of H₂/air polarization curves at the beginning of life (BOL) and after 30,000 cycles for Pt/C baseline vs. Pt/C-([C₂mim]⁺[NTf₂]⁻), Pt/C-([C₄mim]⁺[NTf₂]⁻), and Pt/C-([C₄dmim]⁺[NTf₂]⁻) MEAs, comparison of b) cell potential at 0.8 A cm⁻², c) mass activities obtained at 0.9 V vs. RHE, and d) ECSA for all MEAs at BOL vs. EOL (after 30k cycles). BOL electrochemical properties were obtained after two and three voltage recovery cycles for baseline and IL-containing MEAs, respectively. EOL properties were obtained after one voltage recovery cycle for all MEAs. “

23- p11: "MEA level measurements revealed much higher ECSA losses compared to RDE set-up; as the modified catalysts, Pt/C-([C₂mim]⁺[NTf₂]⁻), Pt/C-([C₄mim]⁺[NTf₂]⁻), Pt/C-([C₄dmim]⁺[NTf₂]⁻) lost 26.8%, 31.1%, 48.3% of the ECSA, respectively. This could be related to the molecular structure of the ILs and the interaction of the cationic alkyl chain with Pt terrace site. In correlation with previously reported observations²⁶, increase in the number of alkyl chains, as well as their length can increase the interaction with Pt sites" Missing in the discussion is, that through the catalyst layer formed also the accessibility Protons towards Pt can be changed through the IL, which would also to another Proton stripping result.

Response:

We restructured the sentences and elaborated more on the differences of proton accessibility between RDE and MEA level measurements.

Change to the manuscript:

“ECSA loss for modified catalysts in MEA-level measurement was 26.8%, 31.1%, 48.3% for Pt/C-([C₂mim]⁺[NTf₂]⁻), Pt/C-([C₄mim]⁺[NTf₂]⁻), and Pt/C-([C₄dmim]⁺[NTf₂]⁻), respectively. As discussed before in the RDE section, this could be related to the molecular structure of the ILs and the interaction of the cationic alkyl chain with Pt terrace site. In correlation with previously reported observations²⁷, increase in the number of alkyl chains, as well as their length can increase the interaction with Pt sites. In addition, MEA level measurements revealed much higher ECSA losses compared to RDE set-up. RDE experiments are carried out in aqueous environment in which all the Pt within the pores is accessible by protons through adsorbed water films. On the other hand, ECSA measurement in MEAs occur at 100% RH where IL molecules are expected to have different interactions with Pt compared to fully aqueous environment.”

Reviewer #3 (Remarks to the Author):

The paper titled 'Ionic Liquids in Polymer Electrolyte Fuel Cells: New State-of-the-art in the Oxygen Reduction Reactivity and Durability' the authors study in length the positive effects of using ionic liquids (IL) as an electrolyte promoting reactant diffusion to the buried catalyst

nanoparticles and in general activity of ORR. In my opinion the paper touches on a very important research in alternative ways to fully utilize low loading PGM catalyst coatings and therefore to make the hydrogen economy a reality. The topic of using IL to fill the normally unutilized support pores in the catalyst layer is relatively new and any information regarding the fundamental processes leading to better utilization is welcomed for further development. The paper is clearly written and the choice of analytical techniques, data analysis and interpretation is sound. Therefore I think that given the novelty and importance of the findings presented in this paper, and the topic in general, the publication of this work in Nature Communication is fully justified. I recommend publishing this paper after considering a few minor comments.

24- Pg2_line48-49 : Authors claim that the carbon support dictates the stability and performance of the catalyst. While certainly the support plays a role in the stability and performance, it does not have a major influence. There is vast literature about the stability and performance drivers of the catalyst layer.

Response:

We agree to reviewer's comment that "dictate" is a strong word for carbon support role in the stability and performance.

Change to the manuscript:

"It has been shown that the carbon support on which the Pt nanoparticles are dispersed plays a role on both stability and performance of the catalyst layer by affecting the porosity, corrosion rate, and mass transport properties¹⁵."

25- Pg5_line134-136: The sentence starting "Difference was seen..." is hard to understand. Do authors mean that for both mesopores and micropores there is the difference compared to the baseline? Or is there also a difference between different ILs?

Response:

Yes, we meant that for micropores there is a huge pore volume difference (about 90%) between baseline and all three IL-containing samples. This means that micropores are fully filled with ILs. For mesopores, this difference is smaller (but still is more than 50%) showing that mesopores are partially filled with IL.

Change to the manuscript:

"For pores between 2 nm to 50 nm, the pore volume decreased at least 50% showing that all IL molecules have partially filled mesopores. On the other hand, over 90% drop in the pore volume is observed for micropores between 1nm and 2nm indicating that the micropores are fully impregnated with ILs. This is in agreement with the significantly reduced nitrogen uptake below 0.2 P/P₀ indicated in Figure 2a.

26- Pg9-10_line210-220: This is interesting. Presumably the IL protects the active sites from oxidizing, improving the activity of the catalyst. But this should be true for all studied IL as they all show hydrophobicity. So why only the C4min show better performance to the Pt baseline? Perhaps, as mentioned later in the paper, this could be due to the IL interaction with the surface? Some ILs also poison the active sites?

Response:

We believe the improvement in the performance stems from the interplay of effects including both IL molecules electrophysical properties and chemical structures. ILs with longer Alkyl chain will have more interaction with Pt terrace sites causing ECSA loss due to blockage of the active sites; however, depending on other properties such as proton conductivity, the remaining active sites will become more active. At the same time, from BET results we observed that the most complex structure has lower packing in the pores helping the oxygen mass transfer to Pt surface.

Therefore, we can conclude that chemical structure should be tuned based on proton conductivity, mass transport resistance, and electrochemical surface area to achieve the optimized performance.

27- Pg10_line219-220: improved compared to what. Baseline?

Change to manuscript:

“Pt/C-([C₄mim]⁺[NTf₂]⁻) catalyst film demonstrated a potential improvement in ORR kinetics as the half-wave potential (E_{1/2}) is slightly shifted towards higher potential compared to the baseline.”

28- Pg11_line256: The authors could mention here the IL/C ratio for the samples used for in-situ characterization.

Response:

In this section, we compared the electrochemical properties of MEAs with optimized IL/C ratio of 1.28.

Change to the manuscript:

“Figure 4, Figure 5 and Table 2 show the electrochemical characteristics of baseline Pt/C and all three IL-modified MEAs with optimized IL/C ratio of 1.28.”

29- Pg13_line314-316: Do not see 100%RH in the plot S7. Can the authors add this for the comparison?

Change to the manuscript:

“We added baseline Pt/C and Pt/C-([C₄mim]⁺[NTf₂]⁻) polarization curves at 100% RH to Figure S7 (HFR-corrected to S8a, and non-HFR corrected to S8b.)”

REVIEWERS' COMMENTS

Reviewer #3 (Remarks to the Author):

In my opinion the authors answered reviewers questions satisfactorily therefore I recommend the paper for publication.

Reviewer #4 (Remarks to the Author):

The authors well address the comments from Reviewers #2 and #3, but only partially addressed those from Reviewer #1, especially the major concerns indicated as comments 1, as well as comments 2 and 3.

1. Regarding comment 1 from Reviewer #1. The research of IL-modified Pt-base catalysts was firstly reported by Snyder et al in 2010 as in Reference 8. A few other researchers have been working in this area and several mechanisms have been explored. The balance between the proton conductivity and oxygen transport resistance was also been investigated by Huang et al and Yan et al (References 29 and 30) by the RDE measurement in liquid electrolytes. The work presented here moves a step forward to study this mechanism in the MEA test. "The authors present a thorough investigation on the structure-performance relationship", and also include more discussion on the mechanisms in the revision. There's novelty with this work, however, the conceptual advance is still not remarkable and not well justified as a paper to be published in Nature Communications.

2. Regarding comment 2. The authors performed an experiment and the results agree well with the reviewer's point. This is really good. However, what is the influence on the mechanisms proposed due to this new understanding of the redepositing of IL molecules with Nafion after drying is not sufficiently discussed.

3. Regarding comment 3 from Reviewer #1. The authors include some revisions to the manuscript, but the question is not answered and the description is still confusing.

Reviewer #4 (Remarks to the Author):

The authors well address the comments from Reviewers #2 and #3, but only partially addressed those from Reviewer #1, especially the major concerns indicated as comments 1, as well as comments 2 and 3.

1. Regarding comment 1 from Reviewer #1. The research of IL-modified Pt-base catalysts was firstly reported by Snyder et al in 2010 as in Reference 8. A few other researchers have been working in this area and several mechanisms have been explored. The balance between the proton conductivity and oxygen transport resistance was also been investigated by Huang et al and Yan et al (References 29 and 30) by the RDE measurement in liquid electrolytes. The work presented here moves a step forward to study this mechanism in the MEA test. "The authors present a thorough investigation on the structure-performance relationship", and also include more discussion on the mechanisms in the revision. There's novelty with this work, however, the conceptual advance is still not remarkable and not well justified as a paper to be published in Nature Communications.

Response:

We Thank the editor and reviewer for giving us the opportunity to highlight the critical achievements of this work.

This work uniquely addresses the role of ionic liquids in complex catalyst layers in relevant membrane electrode assembly environment for the first time. It formulates a clear understanding of ionic liquids role in altering the interactions between ionomer, carbon support, and Pt in the context of catalyst layer morphology.

This work demonstrates that a part of the conclusions obtained previously by rotating disk electrode (RDE) studies remained, as it is mostly aligned with electrostatic interactions, but a broader understanding of hydrophilic/hydrophobic interactions was never achieved in RDE studies. This work establishes the understanding of local and global interactions at triple-phase interfaces and its effect on the performance in practically relevant catalyst in MEA set-up at high currents. None of this has been studied before. MEA testing has become the golden standard to evaluate real-life fuel cell application as RDE measurement is not sufficient to explain fundamental mechanisms because of several limitations including low temperature, liquid electrolyte, low current densities. Understanding these fundamental phenomena in MEA level is absolutely necessary and it brings novelty to the field.

Change to the manuscript: (highlighted in green in the main manuscript)

We made these changes to the manuscript's abstract to highlight the novelty of this work:

"Ionic liquids have shown to be promising additives to the catalyst layer to enhance oxygen reduction reaction in polymer electrolyte fuel cells. However, fundamental understanding of their role in complex catalyst layers in practically relevant membrane electrode assembly environment is needed for rational design of highly durable and active platinum-based catalysts. Here we explore three imidazolium-derived ionic liquids, selected for their high proton conductivity and oxygen solubility, and incorporate them into high surface area carbon black support. Further, we establish a correlation between the physical properties and electrochemical performance of the ionic liquid-modified catalysts by providing direct evidence of ionic liquids role on altering the hydrophilic/hydrophobic interactions within the catalyst layer interface."

Change to the summary:

"Moreover, based on the correlation between in-situ fuel cell testing and ex-situ physical characterization, we established a clear understanding of ILs role in altering the interactions between Pt, IL, and the ionomer at triple-phase interface."

2. Regarding comment 2. The authors performed an experiment and the results agree well with the reviewer's point. This is really good. However, what is the influence on the mechanisms proposed due to this new understanding of the redepositing of IL molecules with Nafion after drying is not sufficiently discussed.

Response:

We thank the reviewer for bringing up this point.

Based on the experiments we performed, we demonstrated 17% to 30% IL loss. The partial re-deposition agreed with Reviewer#1's comments as we addressed in the first revision of the paper; however, the majority of IL is still in the sample, especially within micropores and smaller mesopores (<20 nm). Therefore, there should not be significant change on the conclusion and the mechanism of this paper.

3. Regarding comment 3 from Reviewer #1. The authors include some revisions to the manuscript, but the question is not answered and the description is still confusing.

Response:

In order to clarify this section, we used terms "*local/near-surface hydrophobicity*" and "*global hydrophilicity of the catalyst layer*". The introduction of ILs create a hydrophobic

gradient near the catalyst surface repelling SO_3^- groups. This is called local/near surface hydrophobicity. On the other hand, SO_3^- repulsion to the outer surface will cause global hydrophilicity through the whole catalyst layer. So, one can say the more hydrophobic the IL is, the more the local hydrophobicity induced by it would be.

Changes to the manuscript in green:

“Error! Reference source not found.c shows the time evolution of water mass gain for baseline and IL-modified Pt/C coated films. By comparing the amount of water absorbed by the samples during the first 900s while they are immersed into the water, it can be concluded that baseline Pt/C coated catalyst layer shows the least amount of water uptake of about 18 mg compared to all IL-containing catalyst layers. Higher amount of water adsorption in IL-containing catalyst layers indicates that the global hydrophilicity of Pt/C-IL-Nafion catalyst layers is more than that of the baseline. For instance, Pt/C-([C₄dmim]⁺[NTf₂]⁻)-Nafion catalyst layer has the highest water uptake amount of all (57 mg); hence, it is the most hydrophilic catalyst layer and has the highest global hydrophilicity. Here we can illustrate the potential intramolecular interactions between the IL molecules and Nafion domains with a schematic (Figure S2). IL molecules are prone to create a local hydrophobic gradient near Pt surface and within the smaller mesopores that repels the SO_3^- groups of Nafion to the outer surface resulting in more water absorption to the catalyst layer. Therefore, one can assume that the higher local/near-surface hydrophobicity is, the higher the global hydrophilicity of the whole the catalyst layer along with the Nafion would be. External contact angles shown in Table 1 were calculated using Wilhelmy equation and sudden increase in the water mass gain after each sample was removed from the water (Δmg). The details of experiment can be found in the Materials and Methods section of this paper. Pt/C-([C₄dmim]⁺[NTf₂]⁻)-Nafion film exhibited the smallest external contact angle of 137° and the highest amount of water uptake of 57 mg among all IL-containing samples. This indicates that ([C₄dmim]⁺[NTf₂]⁻) IL molecules create the most local/near-surface hydrophobic microenvironment and induce the strongest electrostatic repulsion on SO_3^- groups of Nafion to reorient to the outer surface. These results are in agreement with zeta potential measurements in which local hydrophobic interactions between the IL molecules and Nafion’s PTFE backbone possibly dictate the conformational structure of the Nafion polymer chains. A stronger

electrostatic repulsion between the IL molecules and Nafion's PTFE backbone results in a higher local charge density at the interface, and larger zeta potential (**Error! Reference source not found.**c and 2d). From tensiometry results, we **observed** that Pt/C-([C₄mim]⁺[NTf₂]⁻)-Nafion catalyst layer **has the largest external contact angel of 148° and the least water uptake of about 33 mg.** **Therefore, Pt/C-([C₄mim]⁺[NTf₂]⁻)-Nafion film** is the most **hydrophobic catalyst layer** compared to the other two **IL-containing catalyst layers.** Hence, ([C₄mim]⁺[NTf₂]⁻) IL **molecules create** the least **local/near-surface** hydrophobic **microenvironment** compared to the other two ILs and **induce weaker** electrostatic repulsion **on** Nafion **chains** confirmed by zeta potential measurements. ”

Supplementary information:

We modified the figure in the supplementary information for clarification.